# CABS: Conflict-Aware and Balanced Sparsification for Enhancing Model Merging

Zongzhen Yang [1 2]  Binhang Qi [1 2 3]  Hailong Sun [1 2 *]  Wenrui Long [1 2]  Ruobing Zhao [1 2]  Xiang Gao [1 2]

## Abstract

Model merging based on task vectors, i.e., the parameter differences between fine-tuned models and a shared base model, provides an efficient way to integrate multiple task-specific models into a multitask model without retraining. Recent works have endeavored to address the conflicts between task vectors, one of the significant challenges faced by model merging, through sparsification; however, two issues significantly limit their performance: *high parameter overlap* and *unbalanced weight distribution*. To address these issues, we propose a simple yet effective framework called CABS (Conflict-Aware and Balanced Sparsification), consisting of **C**onflict-**A**ware Sparsification (CA) and **B**alanced **S**parsification (BS). CA can reduce parameter overlap by applying masks during sequential pruning, ensuring that each task vector retains distinct, non-overlapping parameters. BS leverages $n$:$m$ pruning to preserve critical weights while maintaining an even distribution across layers. Our comprehensive experiments demonstrate that CABS outperforms state-of-the-art methods across diverse tasks and model sizes.

## 1. Introduction

Model merging has gained increasing attention in the deep learning community, particularly in the context of using task vectors for model merging in large language models (LLMs) (Ilharco et al., 2022; Li et al., 2023; Wortsman et al., 2022; Jin et al., 2022; Matena & Raffel, 2022; Singh & Jaggi, 2020; Akiba et al., 2024). This technique has become especially popular for merging homologous models, those derived by fine-tuning the same base model on different tasks, to create a better-performing model. Many of the best-performing models on the LLM leaderboard (Beeching et al., 2023) are built by fine-tuning the base models and subsequently merging them to optimize task-specific performance. Additionally, major enterprises have employed model merging techniques in the development of pre-training models, such as Llama3 (Dubey et al., 2024) and Qwen2 (Yang et al., 2024a; Lu et al., 2024), to enhance generalization capabilities and improve performance across a range of tasks.

Recent studies have further shown that sparsifying task vectors before merging can mitigate parameter conflicts between different task vectors, leading to measurable improvements in merging performance (Yu et al., 2024; Yadav et al., 2024; Davari & Belilovsky, 2023; He et al., 2024). These conflicts can be categorized into two types: (a) conflicts due to redundant parameters, where parameters that contribute little to performance are unnecessarily retained, and (b) conflicts due to overlapping parameters, where task vectors retain parameters that overlap, potentially with significantly different magnitudes or signs. Such overlaps hinder the effectiveness of the merging process.

Sparsifying task vectors, whether selectively or randomly, aims to reduce conflicts in model merging. However, it shares methodological similarities with one-shot pruning, which primarily focuses on model compression. Magnitude-based pruning (Liang et al., 2021) is one of the mainstream pruning techniques, which can estimate the importance of weights and selectively preserve the essential weights, thus being rightfully superior to random pruning. Inspired by pruning techniques, recent model merging studies (Yadav et al., 2024) applied magnitude-based pruning to sparsify task vectors with the important weights retained. However, as pointed out by DARE (Yu et al., 2024), the results are counterintuitive: magnitude-based pruning underperforms compared to random pruning methods in the context of model merging.

Our research explores the reasons behind this discrepancy, especially in a setting where magnitude-based pruning is expected to perform well. Addressing these issues is key to developing high-performance merged models. Specifically,

[1] State Key Laboratory of Complex & Critical Software Environment (CCSE), Beihang University, Beijing, China  [2] Hangzhou Innovation Institute of Beihang University, Hangzhou, China  [3] National University of Singapore, Singapore, Singapore . Correspondence to: Hailong Sun <sunhl@buaa.edu.cn>.

*Proceedings of the 42nd International Conference on Machine Learning*, Vancouver, Canada. PMLR 267, 2025. Copyright 2025 by the author(s).

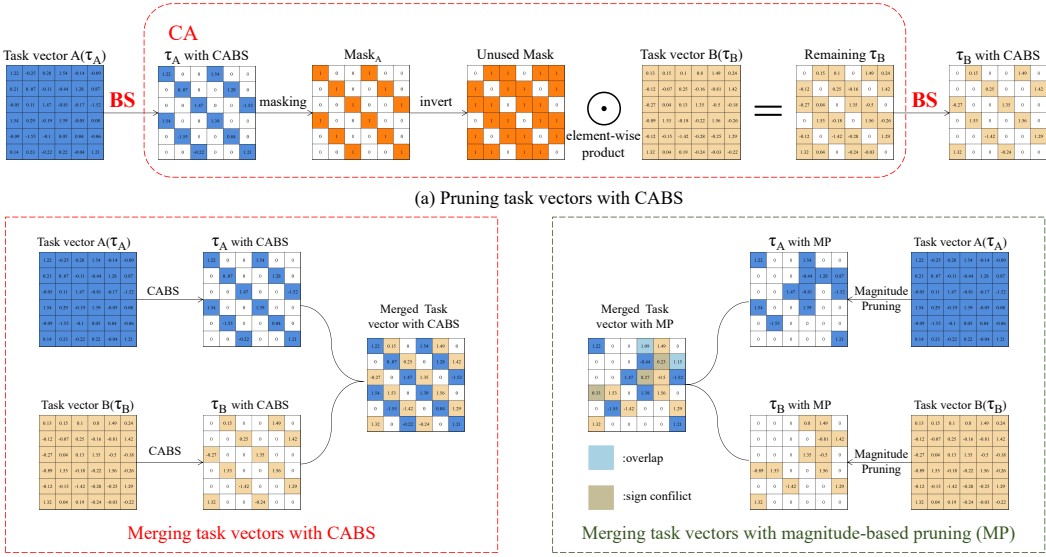

(a) Pruning task vectors with CABS

(b) Merging task vectors with and without CABS

*Figure 1.* Illustration of the **CABS** framework, which enhances model merging by addressing parameter overlap and weight imbalance. By integrating Conflict-Aware Sparsification (CA) and Balanced Sparsification (BS), CABS delivers more effective merging compared to standard merging with magnitude-based pruning (MP), leading to improved model performance.

by analyzing the weight distribution and overlap in task vectors produced by DARE and magnitude-based pruning, we identified two key factors contributing to the underperformance of magnitude-based pruning:

**High Parameter Overlap**: After magnitude-based pruning, the retained weights of different task vectors often exhibit significant overlap, particularly compared to random methods like DARE. The overlap increases conflicts between task vectors during model merging, ultimately degrading the performance of the merged model.

**Unbalanced Weight Distribution**: Magnitude-based pruning tends to distribute retained weights unevenly across the model's weight matrices, with some regions retaining significantly more weights than others. After pruning, the model merging process applies a uniform scaling coefficient globally across the model to restore performance. However, this process amplifies the existing imbalance, ultimately leading to suboptimal performance.

To address the issues above, we propose a novel framework: **Conflict-Aware and Balanced Sparsification (CABS)**. As illustrated in Figure 1, CABS distinguishes itself from existing methods by introducing two key strategies:

**Conflict-Aware (CA) Sparsification**: CA addresses conflicts between task vectors by employing a sequential pruning approach, ensuring *no overlap* between the retained weights of different task vectors. As shown in Figure 1 (a), CA first applies pruning to task vector A (blue, $\tau_A$), and then masks the overlapping weights when pruning task vector B

(yellow, $\tau_B$), resulting in remaining $\tau_B$. This masking technique minimizes conflicts during the merging process by removing shared weights, allowing for more effective task vector merging and improving the final model performance.

**Balanced Sparsification (BS)**: BS addresses the issue of unbalanced weight distribution by applying n:m pruning, which selectively retains $n$ weights out of every $m$ consecutive weights based on magnitude (Zhou et al., 2021). As demonstrated in Figure 1 (a), BS is first applied to $\tau_A$, followed by another application to remaining $\tau_B$ (derived by CA). This ensures a more uniform distribution of weights across layers, reducing the adverse effects of weight concentration in certain regions.

These strategies are effective and easy to implement. We conducted extensive experiments on decoder-based Mistral-7B (Jiang et al., 2023) and encoder-based RoBERTa-Base (Liu, 2019), using tasks from the LLM leaderboard and the GLUE (Wang et al., 2018) dataset. These experiments demonstrate that CABS effectively mitigates the issues caused by magnitude-based pruning. On Mistral-7B, CABS achieved an average performance score of 76.50, surpassing the "ideal" virtual model (76.30), which hypothetically selects the best performance score for each task. CABS also exceeds the state-of-the-art (76.02) and fine-tuned models (75.86). Similarly, on RoBERTa-Base, CABS achieved a score of 81.70, outperforming the SOTA (79.88) by 1.82 points and the vanilla baseline (79.55) by 2.15 points. These results strongly confirm CABS's superiority across diverse neural network architectures and various tasks.

**Our contributions are as follows:**

- We identify two key issues encountered by magnitude-based pruning in the context of task vector sparsification, i.e., high parameter overlap and unbalanced weight distribution.

- We propose the CABS framework, consisting of conflict-aware sparsification and balanced sparsification strategies, which can effectively address the two identified issues.

- We conduct comprehensive experiments across a variety of tasks and model sizes, showing that CABS outperforms state-of-the-art methods.

- We are the first to introduce an "ideal" yet rigorous baseline for evaluation, where CABS outperforms this virtual baseline while all existing methods fall short.

Our code is available at https://github.com/zongzhenyang/CABS.

## 2. Related Work

**Model merging** has become a vital strategy for combining multiple fine-tuned models into a single multitask model without requiring additional training. The simplest merging method is directly averaging the model parameters (Izmailov et al., 2018; Wortsman et al., 2022). However, this naive approach often fails to account for task-specific variations, leading to suboptimal performance. A more refined approach, **Task Arithmetic** (Ilharco et al., 2022), combines task vectors—differences between fine-tuned and pre-trained parameters—using weighted sums controlled by scaling coefficients $\lambda$. These scaling coefficients allow precise control over the contribution of each task vector during merging, playing a critical role in balancing the influence of different tasks. However, it still struggles with parameter redundancy and sign conflicts.

To address these issues, **TIES-Merging** (Yadav et al., 2024) prunes low-magnitude parameters and resolves sign conflicts, reducing interference and preserving critical parameters during merging. **DARE** (Yu et al., 2024), a technique inspired by **Dropout** (Srivastava et al., 2014), reveals the high redundancy in task vectors by randomly dropping 90% of the parameters and rescaling the remaining ones. Using random pruning, DARE has been shown to outperform magnitude-based pruning methods in model merging. However, DARE does not fully explain the reasons for this improvement. Our analysis suggests that DARE helps mitigate some of the overlap and imbalance. However, the random nature of the approach can potentially sacrifice precision.

**Model pruning,** particularly **magnitude pruning** (Zhu & Gupta, 2018), have been extensively studied for their role in optimizing model performance and reducing computational

costs (Liu et al., 2019; Frankle & Carbin, 2018; Gale et al., 2019; Zhu & Gupta, 2018). Magnitude pruning retains parameters based on their magnitude, assuming that larger magnitudes correspond to more critical information (Kovaleva et al., 2021; Puccetti et al., 2022; Yin et al., 2023). However, when applied in the context of model merging, this approach can lead to an unbalanced distribution of retained weights, which exacerbates conflicts during the merging process and results in suboptimal performance.

To address this issue, while **n:m pruning** (Zhou et al., 2021; Xia et al., 2022) was originally designed for pruning and inference acceleration, we discovered that it can be repurposed to control the balance of sparsified task vectors in model merging. Although n:m pruning may not perform as well as unstructured pruning in traditional scenarios, our findings demonstrate that it effectively mitigates weight imbalance, leading to improved performance in merged models.

Our proposed **CABS** method builds upon prior works by introducing CA, a novel approach designed to eliminate parameter overlap during model merging. Additionally, it repurposes the existing n:m pruning technique to mitigate unbalanced weight distribution. Together, CABS effectively enhances the stability and performance of model merging.

## 3. Issues in Task Vector Sparsification for Model Merging

In model merging, particularly when using sparse task vectors to combine models fine-tuned for different tasks, an unexpected phenomenon has emerged: magnitude-based pruning, which typically retains weights with larger absolute values, often underperforms compared to random pruning methods. This result contradicts the intuition that preserving critical knowledge, rather than randomly retaining information, within the task vectors should enhance the performance of the merged model. Our investigation into this phenomenon reveals two key issues: the overlap between retained weights and their unbalanced distribution within each task vector. **High Parameter Overlap.** By comparing the overlap rate between magnitude-based and random pruning methods, our analysis demonstrates that magnitude-based pruning results in a significantly higher parameter overlap between task vectors compared to random pruning methods. As shown in Figure 2, although the overlap rate of magnitude-pruned task vectors decreases gradually with increasing sparsity, it remains significantly higher than that of randomly pruned vectors, especially at higher sparsity levels. This disparity highlights the key issue with magnitude-based pruning, where high overlap persists even as the model becomes sparser.

This elevated overlap in magnitude-pruned vectors introduces conflicts during model merging, as overlapping pa-

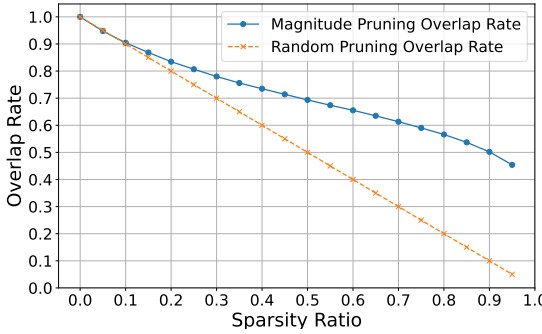

*Figure 2.* The trend of overlap rate along the sparsity ratio shows that the overlap rate achieved by magnitude-based pruning decreases more slowly than that of random pruning, with the gap widening progressively.

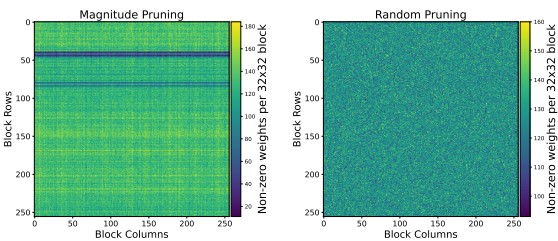

*Figure 3.* Magnitude pruning results in a more concentrated and unbalanced distribution of weights compared to random pruning.

rameters may have significantly different magnitudes or signs between task vectors. For example, if a parameter in task vector $\tau_A$ has a positive value indicating its importance to task A, but the same parameter in $\tau_B$ has a negative value, this sign conflict leads to opposing contributions when merging the two vectors. These conflicts are particularly challenging because they are primarily controlled through scaling coefficients $\lambda$, which serve as key parameters for determining the relative contributions of task vectors during merging. Adjusting $\lambda_A$ for $\tau_A$ can inadvertently affect the contribution of $\tau_B$, reducing the model's ability to perform optimally on individual tasks and ultimately leading to suboptimal task-specific performance. The performance implications of these overlapping parameters are explored in detail in 5.4. For details on how the overlap rate is calculated, please refer to Appendix B.1.

**Unbalanced Weight Distribution.** By visualizing the weight distribution shown in Figure 3, we identified another critical issue: the unbalanced distribution of retained weights caused by magnitude-based pruning. Magnitude pruning often leads to weight concentration in specific regions of the model's weights. This imbalance is further exacerbated by the rescaling process, where certain weights gain disproportionate influence over the model's output, often resulting in suboptimal performance. This uneven dis-

tribution is particularly detrimental after sparsification, as it hampers the merged model's ability to generalize effectively. The performance implications of these unbalanced weights are discussed in detail in 5.4.

To comprehensively analyze this issue, we further examined the weight distributions across different layers of the model, including the query-key-value (QKV) projection and MLP layers, at various sparsity levels (e.g., 50%, 75%, and 90%). These experimental results are provided in Appendix B.2, demonstrating the pervasive nature of the imbalance across different layers and sparsity levels.

## 4. Methodology

To address the aforementioned issues, we propose the CABS (Conflict-Aware and Balanced Sparsification) framework. As illustrated in Figure 1, CABS resolves parameter conflicts and ensures balanced weight distribution, thus enhancing the performance of the merged model. The framework integrates two core strategies: Conflict-Aware Sparsification (CA) and Balanced Sparsification (BS), which will be detailed in the following sections. The detailed implementation of CABS is provided in Appendix B.3.

### 4.1. Conflict-Aware Sparsification (CA)

**Sequential Pruning and Mask Application.** CA aims to eliminate parameter overlap during model merging by employing a sequential pruning strategy. The process begins with the first vector $\tau_A$ being pruned, producing a mask $mask_A$ that marks the positions of the retained weights. This mask is then used to guide the pruning of the second task vector $\tau_B$, ensuring that there is no overlap between the parameters of $\tau_A$ and $\tau_B$.

For the second task vector $\tau_B$, the prior mask $mask_A$ is applied in an inverted form to determine the remaining weights that do not overlap with the first pruned task vector. Specifically, the remaining weights of $\tau_B$ are calculated as:

$$\tau_{B \text{ remaining}} = \tau_B \odot (1 - \text{mask}_A). \tag{1}$$

This ensures that only the non-overlapping weights in $\tau_B$ are retained in the subsequent pruning process. Afterward, a second round of pruning is performed on $\tau_{B \text{ remaining}}$, generating a new sparse mask $mask_B$, which can then be merged with the prior pruned task vector without overlap.

**Minimizing Overlap When Sparsity Limits are Exceeded.** When the sum of the sparsity levels across all task vectors exceeds 1 (e.g., when each vector retains 75% of its parameters), it becomes impossible to achieve zero overlap. In such cases, the objective shifts from eliminating overlap to minimizing it as much as possible. Additional pruning steps are applied selectively to reduce the extent of overlap between task vectors. The detailed implementation

is provided in Appendix B.3.

## 4.2. Balanced Sparsification (BS)

**Block-Based Pruning Strategy**. In BS, the weight matrix is divided into disjoint blocks of $m$ consecutive weights, and within each block, the $n$ weights with the largest absolute magnitude are retained, while the rest are pruned. This strategy is applied uniformly across all layers to ensure a more even weight distribution within each task vector. Minimizing imbalances prevents performance degradation of the merged models. A more detailed discussion about the differences between Balanced Sparsification (BS) and n:m pruning is presented in Appendix B.4.

CABS can be integrated with other model merging techniques, where CA and BS can be applied independently or combined with other approaches to further enhance model merging. Additionally, Our analysis shows that CABS introduces minimal computational and memory overhead compared to standard merging methods, ensuring efficiency and scalability in various model merging scenarios. Detailed analyses are provided in Appendix B.5 and Appendix B.6.

## 4.3. Theoretical Analysis

This section provides a theoretical analysis of how Conflict-Aware Sparsification (CA) reduces parameter overlap, ensures orthogonality of task vectors in parameter space, and mitigates interference during model merging.

**Sparse and Non-Overlapping Task Vectors.** CA employs a sequential pruning strategy to produce sparse task vectors $\tau_A, \tau_B \in \mathbb{R}^{u \times v}$ with non-overlapping parameters. Their binary masks $M_A, M_B \in \{0,1\}^{u \times v}$ satisfy:

$$(M_A)_{ij}(M_B)_{ij} = 0, \quad \forall i,j. \tag{2}$$

The task vectors are defined as:

$$\tau_A = \Delta\mathbf{W}_A \odot M_A, \quad \tau_B = \Delta\mathbf{W}_B \odot M_B. \tag{3}$$

where $\Delta\mathbf{W}_A$, $\Delta\mathbf{W}_B$ are parameter updates from a base model, and $\odot$ denotes elementwise multiplication. This ensures that $\tau_A$ and $\tau_B$ have disjoint non-zero entries. Prior studies (Yu et al., 2024; Yadav et al., 2024) and our experimental results in A.8 confirm that these sparse updates are nearly lossless in retaining task-specific information, as simple rescaling compensates for pruning-induced changes.

**Non-Overlap Implies Orthogonality.** The Frobenius inner product of the task vectors $\tau_A$ and $\tau_B$ is:

$$\langle \tau_A, \tau_B \rangle_F = \sum_{i=1}^{u} \sum_{j=1}^{v} (\tau_A)_{ij}(\tau_B)_{ij}$$
$$= \sum_{i=1}^{u} \sum_{j=1}^{v} (\Delta\mathbf{W}_A)_{ij}(\Delta\mathbf{W}_B)_{ij}(M_A)_{ij}(M_B)_{ij}. \tag{4}$$

Under the non-overlapping condition $(M_A)_{ij}(M_B)_{ij} = 0$, each term in the summation equals zero:

$$(\Delta\mathbf{W}_A)_{ij}(\Delta\mathbf{W}_B)_{ij}(M_A)_{ij}(M_B)_{ij} = 0, \quad \forall i,j. \tag{5}$$

Thus, the inner product reduces to:

$$\langle \tau_A, \tau_B \rangle_F = 0. \tag{6}$$

This guarantees that $\tau_A$ and $\tau_B$ are orthogonal.

**Orthogonality Reduces Interference.** Consider the combined weight update:

$$\Delta\mathbf{W} = \lambda_A \tau_A + \lambda_B \tau_B, \tag{7}$$

where $\lambda_A, \lambda_B \in \mathbb{R}$ are the scaling coefficients for the task vectors. The squared Frobenius norm of the update is:

$$\|\Delta\mathbf{W}\|_F^2 = \|\lambda_A \tau_A\|_F^2 + \|\lambda_B \tau_B\|_F^2 + 2\lambda_A \lambda_B \langle \tau_A, \tau_B \rangle_F. \tag{8}$$

When $\tau_A$ and $\tau_B$ are orthogonal (i.e., $\langle \tau_A, \tau_B \rangle_F = 0$), the cross-term vanishes, and the norm simplifies to:

$$\|\Delta\mathbf{W}\|_F^2 = \|\lambda_A \tau_A\|_F^2 + \|\lambda_B \tau_B\|_F^2. \tag{9}$$

This decoupling ensures that adjusting $\lambda_A$ affects only the contribution of $\tau_A$, with minimal direct interference to $\tau_B$. As a result, task vector contributions can be independently scaled, avoiding interference during model merging.

**On Overlap and Possible Synergy.** While overlap often leads to conflicts, there may be cases where overlapping coordinates have aligned updates, providing synergistic effects. However, identifying exactly which overlap is "helpful" can be challenging, as it requires deep insights into each task's loss surface. Figure 5 shows that excessive overlap typically impairs performance, whereas minimized overlap yields stable and predictable gains. Hence, CA adopts a simpler strategy of systematically limiting overlap, ensuring robust improvements across various tasks.

**Conclusion.** CA eliminates parameter overlap by projecting task vectors onto nearly lossless orthogonal subspaces. Although perfect functional separation cannot be guaranteed in a non-linear neural network, the resulting parameter-space orthogonality ensures that cross-terms vanish during model merging, allowing independent control of each task's contribution through the scaling coefficients ($\lambda$). By minimizing

interference and enabling precise scaling, CA improves both the stability of optimization and the overall efficiency and performance of the merged model. Thus, CA successfully tackles the central challenges of task-vector sparsification, forming a robust foundation for effective model merging.

# 5. Experiments

We conducted extensive experiments to demonstrate the effectiveness of CABS in enhancing performance and stability in model merging across diverse tasks and model scales.

## 5.1. Experimental Setup

**Datasets and Models for Large Language Model Experiments.** For large-scale model evaluation, we utilized the LLM Leaderboard benchmark, encompassing six key tasks: AI2 Reasoning Challenge (Clark et al., 2018), HellaSwag (Zellers et al., 2019), MMLU (Hendrycks et al., 2020), TruthfulQA (Lin et al., 2022), Winogrande (Sakaguchi et al., 2021), and GSM8K (Cobbe et al., 2021). These tasks were assessed using the Eleuther AI Language Model Evaluation Harness (Gao et al., 2024), a standardized framework designed to test generative language models across various tasks. The models used in our experiments were based on the Mistral-7b-v0.1 backbone and included fine-tuned variants such as WildMarcoroni-Variant1-7B and WestSeverus-7B-DPO-v2.

In addition, we conducted a new set of experiments using the Open LLM Leaderboard 2 (Fourrier et al., 2024), which includes six tasks: IFEval (Zhou et al., 2023), BBH (Suzgun et al., 2022), MATH (Hendrycks et al., 2021), GPQA (Rein et al., 2023), MUSR (Sprague et al., 2024), and MMLU-PRO (Wang et al., 2024). For these experiments, we employed the qwen-2.5-7b-instruct (Yang et al., 2024b) model as the backbone and evaluated fine-tuned fq2.5-7b-it and Tsunami-0.5-7B-Instruct to assess performance across these additional benchmarks. More details about the datasets and models are provided in Appendix B.7.

**Datasets and Models for Small Language Model Experiments.** For evaluating small-scale models, we utilized the GLUE benchmark, which includes four binary classification tasks: CoLA (Warstadt et al., 2019), SST-2 (Socher et al., 2013), MRPC (Dolan & Brockett, 2005), and RTE (Dagan et al., 2005; Bar-Haim et al., 2006; Giampiccolo et al., 2007; Bentivogli et al., 2009). To increase task difficulty and diversity, we also included the multiple-choice reading comprehension task RACE (Lai et al., 2017) and the question-answering task SQuAD (Rajpurkar, 2016). We utilized RoBERTa (Liu, 2019) and GPT-2 (Radford et al., 2019) as pre-trained backbones, with fine-tuned models sourced from HuggingFace. Due to the unavailability of test labels, the original validation sets were repurposed as test

sets. Additional details are provided in Appendix B.8.

**Evaluation Metrics.** Performance was evaluated primarily using accuracy for GLUE tasks. For tasks from the LLM Leaderboard, we used task-specific metrics, such as success rates and accuracy, depending on the default evaluation metric for each task. Detailed explanations of the evaluation metrics and the rationale behind these choices can be found in Appendix B.9.

**Baselines.** We compared CABS against several baseline methods in two main categories: conflict handling and sparsification strategies. For conflict handling, we evaluated Task Arithmetic (Ilharco et al., 2022) and TIES-Merging (Yadav et al., 2024). For sparsification, we compared CABS with DARE (Yu et al., 2024), Magnitude Pruning (Zhu & Gupta, 2018), SparseGPT (Frantar & Alistarh, 2023), and Wanda (Sun et al., 2023).

It is worth mentioning that, to assess how far current model merging methods are from the ideal performance expected in this research field, we introduce an **"ideal model"** as a strict and meaningful baseline. The ideal model represents a hypothetical scenario where the merged model achieves optimal performance for each task. This baseline is constructed by selecting the best-performing individual task-specific model for each task, providing an upper bound for comparison.

**Other Implementation Details.** Details on the grid search strategy and exact values of $\lambda$ are provided in Appendices B.10 and B.11, respectively. Hardware setups, evaluation strategies, and hyperparameter configurations are detailed in Appendix B.12.

## 5.2. Performance of CABS on Small LMs

We conducted experiments on three task sets to evaluate the effectiveness of CABS in merging small-scale models (e.g., RoBERTa): 1) 2-task set comprising RTE and MRPC, 2) 4-task set comprising RTE, CoLA, MRPC, and SST-2, and 3) 6-task set comprising RTE, CoLA, MRPC, SST-2, RACE, and SQuAD.

**Overall Performance.** Table 1 presents the performance for merging four task vectors. Among the baselines, "Task Arithmetic" represents a vanilla approach without pruning, while other methods incorporate pruning techniques. For our proposed CABS, the last four rows display results with different orders of sequential pruning (e.g., "MRSC" indicates pruning task vectors of MRPC, RTE, SST-2, and CoLA sequentially). The last column displays the overall performance of the merged model (i.e., the average result across four tasks), with the results in brackets indicating the improvement over Task Arithmetic.

As we can see, random-based pruning methods offer limited performance improvements (e.g., "TIES-Merging + DARE"

Table 1. Performance of merging four task vectors (sparsity=0.90).

| Method | CoLA | SST-2 | RTE | MRPC | Avg |
|---|---|---|---|---|---|
| Ideal Model | 85.04 | 94.04 | 79.42 | 91.18 | 87.42 |
| Task Arithmetic | 76.32 | 90.83 | 69.68 | 81.37 | 79.55 |
| + Magnitude | 82.07 | 87.04 | 65.34 | 79.66 | 78.53 (-1.02) |
| + DARE | 76.99 | 90.14 | 70.76 | 81.13 | 79.76 (+0.21) |
| TIES-Merging | 82.36 | 86.93 | 61.01 | 79.41 | 77.43 (-2.12) |
| + DARE | 77.66 | 90.94 | 69.31 | 81.62 | 79.88 (+0.33) |
| CABS (CSRM) (Ours) | 78.24 | **92.32** | 74.37 | 81.62 | 81.64 (+2.09) |
| CABS (SCMR) (Ours) | **78.52** | 91.97 | 73.65 | 82.60 | 81.69 (+2.14) |
| CABS (RCMS) (Ours) | 77.76 | 92.09 | **75.09** | 81.62 | 81.64 (+2.09) |
| CABS (MRSC) (Ours) | 76.89 | 92.09 | 74.73 | **83.09** | **81.70 (+2.15)** |

Table 2. Impact of task number on model merging performance.

| Method | 2 tasks | 4 tasks | 6 tasks |
|---|---|---|---|
| Ideal Model | 85.30 | 87.42 | 83.54 |
| Task Arithmetic | 80.15 | 79.55 | 66.56 |
| + Magnitude | 80.38 (+0.23) | 78.53 (-1.02) | 68.28 (+1.72) |
| + DARE | 80.58 (+0.43) | 79.76 (+0.21) | 67.23 (+0.67) |
| TIES-Merging | 80.20 (+0.05) | 77.43 (-2.12) | 65.46 (-1.10) |
| + DARE | 80.65 (+0.50) | 79.88 (+0.33) | 66.95 (+0.39) |
| **CABS (Ours)** | **81.49 (+1.34)** | **81.70 (+2.15)** | **69.62 (+3.06)** |

improves by only 0.33). Magnitude-based pruning even degrades performance, consistent with previous findings. CABS achieves the highest average accuracy of 81.70, surpassing Task Arithmetic by 2.15 and delivering substantial improvements over all other methods. Additionally, the pruning order can affect the performance of the merged model on specific tasks. For instance, the best results for CoLA (78.52) and SST-2 (92.32) are achieved when these tasks are pruned first. However, the variation has minimal impact on overall performance. On average, all pruning orders achieve comparable results (81.64 to 81.70), highlighting the robustness of CABS in handling variations in pruning order despite task-specific differences.

**Performance Impact of Number of Tasks.** Table 2 highlights the performance impact of task number on model merging. As the number of tasks increases, overall merging performance declines due to the increasing heterogeneity of tasks. This effect is particularly evident when transitioning from 4 to 6 tasks, as including QA and multiple-choice tasks (RACE and SQuAD) introduces additional complexity.

Despite these challenges, CABS consistently outperforms baseline methods across all scenarios. Compared to Task Arithmetic, CABS achieves improvements of 1.34, 2.15, and 3.06 for 2-task, 4-task, and 6-task sets, respectively. These results highlight the robustness and scalability of CABS in handling diverse and complex task sets, maintaining significant gains even as task heterogeneity increases.

Table 3. Performance comparison on LLM Leaderboard using different methods (sparsity=0.75).

| Method | ARC | Hella. | MMLU | TQA | Wino. | GSM8K | AVG |
|---|---|---|---|---|---|---|---|
| WestSeverus | 71.30 | 88.26 | 63.92 | 72.72 | 83.69 | 74.27 | 75.69 |
| WildMarcoroni | 73.63 | 88.67 | 63.96 | 70.07 | 84.34 | 74.48 | 75.86 |
| Ideal Model | 73.63 | 88.67 | 63.96 | 72.72 | 84.34 | 74.48 | 76.30 |
| Task Arithmetic | 72.52 | 89.25 | 63.39 | 74.00 | 83.46 | 73.38 | 76.02(-0.28) |
| + Magnitude | 71.93 | **89.32** | 63.18 | 73.85 | 84.12 | 72.22 | 75.77(-0.53) |
| + DARE | 72.64 | 88.86 | 63.54 | 72.82 | 84.03 | 73.44 | 75.89(-0.41) |
| TIES-Merging | 71.42 | 89.17 | 63.16 | 73.82 | **84.74** | 73.01 | 75.89(-0.41) |
| + DARE | 71.87 | 88.95 | **63.56** | 72.87 | 84.61 | 73.21 | 75.85(-0.46) |
| **CABS (Ours)** | 72.92 | 88.89 | 63.50 | **74.41** | 84.63 | **74.65** | **76.50(+0.20)** |

The detailed results for each configuration are presented in Table 1, Table 9, and Table 10. Additional results for the CoLA and SST-2 tasks can be found in Table 11 (Appendix A.3), and the results for the GPT-2 model are provided in Table 12 (Appendix A.4).

### 5.3. Performance of CABS on Large LMs

**Overall Performance.** Table 3 shows the results on large LMs. The last column, "AVG", represents the average performance of merged models across six tasks, with the numbers in parentheses indicating the gap from the "ideal model". Existing methods, whether based on magnitude pruning or random pruning, show similar performance and fail to outperform Task Arithmetic. These baselines remain notably below the "ideal model", highlighting the challenge of surpassing this strict baseline. In contrast, CABS achieves an average score of 76.50, surpassing all baselines and even exceeding the "ideal model".

The result highlights the advantage of model merging in enhancing generalization. While the merged model may not surpass the "ideal model" on every individual task, it often achieves superior performance on specific tasks. For example, in the TruthfulQA task (see column "TQA" in Table 3), the fine-tuned models scored 72.72 and 70.07, while the vanilla baseline reached 74.00, and CABS further increases the score to 74.41. Overall, CABS achieved an average performance of 76.50, exceeding the "ideal model" and significantly outperforming the best baseline score of 76.02. The result underscores the effectiveness of CABS in model merging for large-scale models.

**Notable Achievement on Open LLM Leaderboard 2.** As of February 24, 2025, our CABS framework enabled the creation of four merged models (qwen2.5-7b-cabs v0.1 through v0.4), which dominated the **top four** positions among models with 8B parameters or fewer on the Open LLM Leaderboard, As shown in Table 4. this achievement underscores CABS' effectiveness in improving model performance.

**Performance Impact of Sparsity Rate.** Figure 4 illus-

*Table 4.* Results of 7B LLMs on the Open LLM Leaderboard 2(sparsity=0.75).

| Models | IFEval | BBH | MATH | GPQA | MUSR | MMLU | AVG |
|---|---|---|---|---|---|---|---|
| Tsunami-0.5-7b | 74.00 | 36.14 | 50.45 | 7.83 | 12.21 | 37.92 | 36.43 |
| fq2.5-7b | 73.99 | **36.36** | 46.22 | 6.94 | 17.54 | 37.92 | 36.50 |
| cabs-v0.1(Ours) | 75.06 | 35.84 | 47.96 | 8.50 | 14.17 | 37.84 | 36.56 |
| cabs-v0.2(Ours) | 74.18 | 36.28 | 49.02 | 7.61 | 14.86 | 37.75 | 36.61 |
| cabs-v0.3(Ours) | 75.70 | 35.96 | 49.32 | 7.61 | 15.24 | 37.80 | 36.94 |
| cabs-v0.4(Ours) | **75.83** | **36.36** | 48.49 | 7.72 | 15.17 | 37.73 | 36.88 |

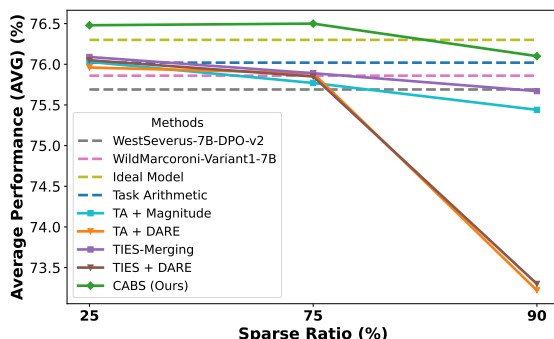

*Figure 4.* Performance comparison across sparsity.

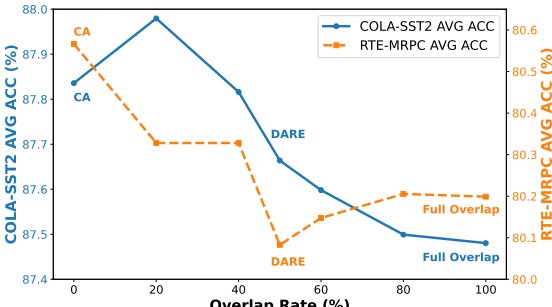

*Figure 5.* Merged model performance decreases as overlap rate increases, underscoring the importance of CA in reducing conflicts.

trates the performance of different model merging methods across varying sparsity levels. The dashed lines represent the performance of the two pre-trained models, the merged model obtained via Task Arithmetic, and the ideal model. The solid lines indicate the performance of merged models obtained using different methods at varying sparsity levels, highlighting their trends as sparsity increases.

As sparsity increases, all methods experience a performance decline, with the limitations of existing methods becoming particularly pronounced at 90% sparsity. Random pruning-based methods (e.g., "TA + DARE") suffer the most significant degradation due to the loss of critical weights, while magnitude-based pruning approaches (e.g., "TA + Magnitude") also underperform due to imbalanced weight distribution. In contrast, CABS consistently achieves superior performance across all sparsity levels, demonstrating its robustness and ability to preserve essential information even under high sparsity constraints. More detailed results and discussions for each sparsity level are presented in Table 3, Table 13, and Table 14.

### 5.4. Ablation Studies and Discussion

Within the CABS framework, we first analyze the independent contributions of CA and BS by examining the impact of parameter overlap and unbalanced weight distribution on model merging. Next, we perform ablation studies to isolate the contributions of CA and BS, demonstrating the importance of both strategies for achieving optimal results.

**Performance Impact of Overlap Rate (CA's Contribution).** We examined the impact of varying overlap rates on merged model performance to validate the importance of CA. The experiment was conducted on two task pairs (RTE-MRPC and CoLA-SST2) at a fixed sparsity level of 0.50, using random pruning for fair comparison. To achieve the target overlap rate ranging from 0% (no overlap, i.e., CA) to 100% (full overlap), we first pruned one task vector, then adjusted the pruning of the second vector by controlling the ratio of retained weights in the overlapping and non-overlapping regions.

As shown in Figure 5, a lower overlap rate generally leads to better performance. Notably, the 50% overlap rate, which corresponds to the expected overlap rate of DARE, performs worse than the non-overlapping condition achieved by CA. This result highlights the importance of minimizing parameter overlap, as achieved by CA.

**Comparisons with Magnitude-Based and Advanced Pruning Methods (BS's Contribution).** Table 5 compares BS to magnitude-based pruning approaches (including layer-wise and row-wise) and advanced pruning methods (i.e., SparseGPT and WANDA). The results show a clear progression in performance as balance improves: layer-wise pruning achieves 80.38, row-wise pruning improves to 80.61, and BS further increases to 81.30. This demonstrates that enhancing weight distribution balance can contribute to better model merging performance.

Advanced pruning methods, while effective in traditional pruning tasks, perform similarly to the worst-performing layer-wise magnitude pruning (e.g., 80.34 for SparseGPT). This indicates that such methods are less suitable for task vector sparsification in model merging scenarios. By effectively addressing weight distribution imbalances, BS demonstrates its robustness and effectiveness in improving model merging performance.

Table 5. Comparison of sparsity strategies (sparsity=0.9).

| Method | RTE | MRPC | AVG |
|---|---|---|---|
| Fine-tuned on RTE | 79.42 | 25.98 | 52.70 |
| Fine-tuned on MRPC | 47.29 | 91.18 | 69.24 |
| Task Arithmetic | 73.29 | 87.01 | 80.15 |
| + DARE | 72.92 | **88.24** | 80.58(+0.43) |
| + Magnitude (layer-wise) | **74.73** | 86.03 | 80.38 (+0.23) |
| + Magnitude (row-wise) | 74.06 | 87.05 | 80.61 (+0.46) |
| + SparseGPT | 72.92 | 87.75 | 80.34 (+0.19) |
| + WANDA | 73.29 | 87.50 | 80.40 (+0.25) |
| **BS (Ours)** | 74.37 | 88.23 | **81.30 (+1.08)** |

Table 6. Ablation study of CABS across different sparsity levels.

| Sparsity Level | Method | Overlap Rate | Avg Accuracy |
|---|---|---|---|
| 0% | Task Arithmetic | 100.00 | 76.02 |
| 25% | TA+magnitude | 80.69 | 76.03 |
|  | CA Only | 66.67 | 76.29 |
|  | BS Only | 80.97 | 76.33 |
|  | CABS | 66.67 | 76.48 |
| 75% | TA+magnitude | 71.42 | 75.77 |
|  | CA Only | 0.00 | 76.21 |
|  | BS Only | 58.63 | 76.24 |
|  | CABS | 0.00 | 76.50 |

**Combined Effect of CA and BS.** To validate the effectiveness of CA and BS, we conducted an ablation study comparing configurations with only CA, only BS, and the full CABS framework. As shown in Table 6, CABS not only benefits from CA and BS independently improving performance, but their combination also minimizes overlap across all sparsity levels and achieves the highest accuracy.

In conclusion, our ablation studies confirm the necessity of reducing overlap rates and maintaining balanced weight distribution for optimal model merging. They validate the crucial roles of CA and BS, showing that combining both strategies achieves the best performance across various tasks and sparsity settings. Furthermore, we performed a series of analyses on varying $n : m$ ratios and provided additional results on the impact of different pruning orders in Appendix A.6 and A.7. These results further demonstrate the robustness of the CABS framework. Additionally, we conducted rescaling experiments and found that applying rescaling to magnitude-pruned task vectors can restore performance to levels comparable to the original models, similar to what has been observed with DARE's random pruning method. Detailed results of these rescale experiments are included in Appendix A.8.

### 5.5. Limitations and Future Work

**General Limitations.** Like other task vector-based methods, our approach is limited to models with identical architectures due to the element-wise operations used in merging model weights. This constraint restricts the generalization of the framework to models with homogeneous structures. Furthermore, reliance on manual adjustment of the parameter $\lambda$ remains a common challenge, especially for large-language models, which requires trial and error to optimize model performance.

**Limitations Specific to CABS.** CABS introduces two new hyperparameters—the sparse sequence and the n:m ratios—unique to its design, as discussed in Appendix A.7 and A.6. While these hyperparameters were not particularly sensitive in our experiments, they add complexity and increase computational cost.

**Future Work.** Several directions could help overcome these limitations. Expanding model merging techniques to include heterogeneous architectures represents a key area for future research. Additionally, improving the performance of merged models in multi-task settings—where current approaches do not yet match the performance of original single-task models—remains a priority.

### 6. Conclusion

In this work, we revealed two issues in model merging: high parameter overlap and unbalanced weight distribution in task vector sparsification. To address these issues, we proposed Conflict-Aware and Balanced Sparsification (CABS). CABS effectively reduces overlap and ensures a balanced distribution of retained weights, thus enhancing model merging across various tasks and model sizes. Extensive experiments on both small- and large-scale models demonstrated CABS's effectiveness in improving merged models' performance and generalization.

### Impact Statement

This paper presents work whose goal is to advance the field of Machine Learning. There are many potential societal consequences of our work, none which we feel must be specifically highlighted here.

### Acknowledge

This work was supported by National Key Research and Development Program of China under Grant No. 2024YFB3309602, National Natural Science Foundation of China under Grant No.62472017, and Guangxi Collaborative Innovation Center of Multi-source Information Integration and Intelligent Processing.

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

# A. Additional Experiments Results

## A.1. Impact of Lambda Search Grid on Performance

In this section, we analyze the impact of different lambda search grids on the performance of various model merging methods. Our experiments demonstrate the importance of using fine-grained grid intervals to fairly compare the effectiveness of these methods. Table 7 provides results across different grid intervals (0.01, 0.05, and 0.1) for several methods.

For most methods, performance declines as the grid interval increases, underscoring the importance of finer grids to accurately capture optimal lambda values. Coarser grids often miss these values, leading to noticeable drops in performance.

Interestingly, the DARE method maintains stable performance even with coarser grids (0.05 and 0.1). This is because the optimal lambda for DARE happens to coincide with a multiple of 0.1, resulting in no significant performance loss with coarser grids. However, when we exclude such coincidental "sweet spot" lambdas, as shown in Table 8, DARE also exhibits a significant performance drop. This observation reinforces the idea that fine grid intervals are crucial for a fair and thorough evaluation of all methods. A finer grid ensures that all methods have an equal opportunity to find the best-performing lambda, though this must be balanced with computational cost

On the other hand, the CABS method demonstrates robust performance across all grid intervals. It consistently outperforms other methods, and its relative insensitivity to grid coarseness suggests that CABS is more robust and reliable under varying hyperparameter settings. This robustness, combined with its superior performance, makes CABS a strong choice for model merging.

*Table 7.* Performance comparison across different lambda grid intervals."TA" means "Task Arithmetic"

| Grid Interval | Task Arithmetic | DARE | TA-Magnitude | TIES-DARE | TIES-Merging | CABS |
|---|---|---|---|---|---|---|
| 0.01 | 80.15 | 80.58(+0.43) | 80.38(+0.23) | 80.65(+0.40) | 80.20(+0.05) | **81.49(+0.91)** |
| 0.05 | 79.85 | 80.58(+0.73) | 79.90(+0.05) | 79.91(+0.06) | 79.84(-0.01) | **81.19(+1.34)** |
| 0.10 | 79.43 | 80.58(+1.15) | 79.66(+0.23) | 79.14(-0.29) | 79.83(+0.40) | **80.82(+1.39)** |

*Table 8.* Performance comparison across different lambda grid intervals excluding one pair sweet spot lambdas in DARE.

| Grid Interval | Task Arithmetic | DARE | TA-Magnitude | TIES-DARE | TIES-Merging | CABS |
|---|---|---|---|---|---|---|
| 0.01 | 80.15 | 80.58(+0.43) | 80.38(+0.23) | 80.65(+0.40) | 80.20(+0.05) | **81.49(+0.91)** |
| 0.05 | 79.85 | 79.44(-0.41) | 79.90(+0.05) | 79.91(+0.06) | 79.84(-0.01) | **81.19(+1.34)** |
| 0.10 | 79.43 | 78.55(-0.88) | 79.66(+0.23) | 79.14(-0.29) | 79.83(+0.40) | **80.82(+1.39)** |

## A.2. Detailed results of CABS on Small LMs Merging

This section provides Detailed results for the experiments on small LMs merging in Table 2. Table 9 compares the performance on the RTE-MRPC task pair at 90% sparsity, showing that CABS outperforms all baselines, achieving the highest average score of 81.49 (+1.34). Similarly, Table 10 presents the results of merging six task vectors at the same sparsity level, where CABS also demonstrates superior performance with an average score of 69.62 (+3.06), significantly surpassing other methods. These results highlight the effectiveness of CABS in achieving robust and consistent improvements across multiple tasks, even under high sparsity constraints.

## A.3. Additional Experiments on other Task Pairs for Small-Scale Experiments

In this section, we present additional results for the CoLA-SST2 task pair to complement the main text's findings on RTE and MRPC. These tasks were selected to further validate the robustness and effectiveness of the proposed **CABS** method across different types of natural language processing tasks, particularly focusing on tasks involving linguistic acceptability and sentiment analysis.

Table 11 provides a detailed comparison of various model merging methods on the CoLA and SST2 tasks. The **CABS**

*Table 9.* Performance comparison on RTE-MRPC task pair using different methods (sparsity=0.9).

| Method | RTE | MRPC | AVG |
|---|---|---|---|
| Fine-tuned on RTE | 79.42 | 25.98 | 52.70 |
| Fine-tuned on MRPC | 47.29 | 91.18 | 69.24 |
| Task Arithmetic | 73.29 | 87.01 | 80.15 |
| + Magnitude | **74.73** | 86.03 | 80.38(+0.23) |
| + DARE | 72.92 | 88.24 | 80.58(+0.43) |
| TIES-Merging | 74.37 | 86.03 | 80.20(+0.05) |
| + DARE | 72.56 | 88.73 | 80.65(+0.50) |
| **CABS (Ours)** | 74.01 | **88.97** | **81.49(+1.34)** |

*Table 10.* Performance comparison of merging six task vectors(sparsity=0.9).

| METHOD | RTE | MRPC | CoLA | SST2 | RACE | SQuAD | AVG |
|---|---|---|---|---|---|---|---|
| Ideal Model | 79.42 | 91.18 | 85.04 | 94.04 | 71.71 | 79.82 | 83.54 |
| Task Arithmetic | 67.15 | 79.41 | 72.00 | 85.78 | 56.21 | 38.82 | 66.56 |
| + Magnitude | 72.56 | 81.13 | 75.26 | 87.50 | 56.99 | 36.23 | 68.28 (+1.72) |
| + DARE | 71.12 | 65.44 | 72.48 | 83.37 | 59.57 | 51.39 | 67.23 (+0.67) |
| TIES-Merging | 68.94 | 86.01 | 66.43 | 83.33 | 40.11 | 47.94 | 65.46 (-1.10) |
| + DARE | **74.40** | 83.83 | 72.92 | 56.37 | **60.38** | **53.80** | 66.95 (+0.39) |
| CABS(Ours) | 68.95 | 82.11 | 73.92 | **90.83** | 58.97 | 42.96 | **69.62 (+3.06)** |

method demonstrates superior performance, achieving the highest average scores across both tasks. The normalized accuracy scores (COLA-N and SST2-N) further emphasize the effectiveness of the **CABS** method, showing consistent improvements over the baseline methods.

The modest gains observed in the CoLA-SST2 experiments, similar to those in the RTE-MRPC pair, can be attributed to the fine-grained lambda grid search. This search process, which fine-tunes the sparsification parameters, improves the overall performance across all methods, thereby reducing the performance gaps. However, **CABS** still outperforms other methods, indicating its robustness in handling task-specific nuances during model merging.

*Table 11.* Performance comparison on COLA-SST2 task pair using different methods.(sparsity=0.9)

| METHOD | COLA | SST2 | AVG | COLA-N | SST2-N | AVG-N |
|---|---|---|---|---|---|---|
| Fine-tuned model on COLA | 85.04 | 50.92 | 67.98 | 100.00 | 54.15 | 77.08 |
| Fine-tuned model on SST2 | 68.74 | 94.04 | 81.39 | 80.83 | 100.00 | 90.32 |
| Task Arithmetic | 81.59 | 92.89 | 87.24 | 95.94 | 98.78 | 97.36 |
| Task Arithmetic + Magnitude | 81.69 | 93.46 | 87.58(+0.34) | 96.06 | 99.38 | 97.72(+0.36) |
| Task Arithmetic + DARE | 81.78 | 93.46 | 87.62(+0.38) | 96.17 | 99.38 | 97.78(+0.42) |
| TIES-Merging | 81.21 | 93.58 | 87.40(+0.16) | 95.5 | 99.51 | 97.51(+0.19) |
| TIES-Merging + DARE | 81.78 | **93.69** | 87.74(+0.50) | 96.17 | **99.63** | 97.90(+0.54) |
| **CABS (Ours)** | **82.55** | 93.35 | **87.95(+0.71)** | **97.07** | 99.27 | **98.17(+0.81)** |

The results from these additional experiments support the conclusions drawn in the main paper, highlighting **CABS** as a robust and effective model merging technique across various tasks and evaluation metrics.

### A.4. Additional Experiments on GPT-2-Based Models

we have also extended our experiments to include other architectures, specifically GPT-2-based models (Radford et al., 2019). The results, summarized in Table 12, highlight the performance of CABS and other methods on tasks derived from **FusionBench** (Tang et al., 2024).

The results demonstrate that **CABS** outperforms all other methods and is the only method to surpass the Ideal Model.

Table 12. Performance comparison on GPT-2-based models.

| Method | CoLA | MRPC | AVG |
|---|---|---|---|
| Fine-tuned on CoLA | 76.80 | 68.40 | 72.60 |
| Fine-tuned on MRPC | 30.80 | 80.39 | 55.60 |
| Ideal Model | 76.80 | 80.39 | 78.60 |
| Task Arithmetic (Dense) | 75.55 | 77.45 | 76.50 (-2.10) |
| TA + DARE | 76.70 | 77.21 | 76.95 (-1.65) |
| TA + Magnitude | 76.61 | 79.66 | 78.13 (-0.47) |
| TIES + DARE | 77.09 | 76.72 | 76.91 (-1.69) |
| TIES-Merging | 76.89 | 77.94 | 77.42 (-1.18) |
| **CABS (Ours)** | **76.41** | **80.88** | **78.65 (+0.05)** |

Table 13. Performance comparison on LLM Leaderboard using different methods. (sparsity=0.25)

| Method | ARC | Hella. | MMLU | TQA | Wino. | GSM8K | AVG |
|---|---|---|---|---|---|---|---|
| WestSeverus-7B-DPO-v2 | 71.30 | 88.26 | 63.92 | 72.72 | 83.69 | 74.27 | 75.69 |
| WildMarcoroni-Variant1-7B | 73.63 | 88.67 | 63.96 | 70.07 | 84.34 | 74.48 | 75.86 |
| ideal model | 73.63 | 88.67 | 63.96 | 72.72 | 84.34 | 74.48 | 76.30 |
| Task Arithmetic | 72.52 | 89.25 | 63.39 | 74.00 | 83.46 | 73.38 | 76.02(-0.28) |
| + Magnitude | 71.67 | 89.15 | 63.42 | 74.05 | 84.37 | 73.53 | 76.03(-0.27) |
| + DARE | 72.30 | 88.77 | **63.84** | 72.08 | 84.40 | 74.40 | 75.96(-0.34) |
| TIES-Merging | 72.41 | **89.34** | 63.40 | 74.03 | 83.64 | 73.69 | 76.09(-0.21) |
| + DARE | 72.30 | 88.63 | 63.76 | 72.16 | 85.06 | 74.37 | 76.05(-0.25) |
| TIES-Merging + CABS | **72.97** | 89.20 | 63.46 | 74.00 | **85.16** | 74.50 | 76.44(+0.14) |
| **CABS (Ours)** | 72.75 | 89.17 | 63.48 | **74.08** | 84.66 | **74.73** | **76.48(+0.18)** |

Although the improvement margin is relatively smaller due to the upper-bound constraint imposed by the Ideal Model, CABS consistently proves its effectiveness across tasks.

Interestingly, magnitude pruning shows unexpectedly strong results on GPT-2-based models, surpassing DARE by a significant margin. This contrasts with previous experiments on other architectures, suggesting a potential architecture-specific behavior in existing pruning methods. Nevertheless, CABS maintains its advantages across different architectures, showcasing its robustness and adaptability.These findings underscore the versatility of CABS and its potential for diverse architectures.

### A.5. Detailed results of CABS on Large LMs Merging

This section provides detailed results for the experiments on large LMs merging under different sparsity levels. Table 13 presents the results at 25% sparsity. CABS achieves the highest average score of 76.48 (+0.18), outperforming all baselines and closely approaching the ideal model's performance. The results demonstrate the robustness of CABS in preserving task-relevant information and mitigating performance degradation, even under moderate sparsity constraints.

Table 14 shows the results at a much higher sparsity level of 90%. Despite the challenging conditions, CABS maintains competitive performance with an average score of 76.10 (-0.20), surpassing other methods, including Task Arithmetic, TA-dare, and Ties-magnitude. These results highlight the effectiveness of CABS in achieving stable and high-quality model merging, even at extreme sparsity levels.

### A.6. Effect of Different n:m Ratios at Fixed Sparsity Levels

This section examines how different n:m ratios impact the performance of the merged model while keeping the overall sparsity fixed at 75%. The results in Table 15 indicate that while higher n:m ratios (e.g., 64:256) tend to show slight improvements, the overall impact of varying n:m ratios remains relatively subtle, suggesting that model performance is not highly sensitive to these values.

Table 14. Performance comparison on LLM Leaderboard using different methods. (sparsity=0.90)

| METHOD | ARC | Hella. | MMLU | TQA | Wino. | GSM8K | AVG |
|---|---|---|---|---|---|---|---|
| Mistral-7B-v0.1 | 59.98 | 83.31 | 64.16 | 42.15 | 78.37 | 37.83 | 60.97 |
| WestSeverus-7B-DPO-v2 | 71.30 | 88.26 | 63.92 | 72.72 | 83.69 | 74.27 | 75.69 |
| WildMarcoroni-Variant1-7B | 73.63 | 88.67 | 63.96 | 70.07 | 84.34 | 74.48 | 75.86 |
| Ideal Model | 73.63 | 88.67 | 63.96 | 72.72 | 84.34 | 74.48 | 76.30 |
| Task Arithmetic (Dense) | 72.52 | 89.25 | 63.39 | 74.00 | 83.46 | 73.38 | 76.02 |
| TA-dare | 70.73 | 87.18 | 60.15 | 70.69 | 82.64 | 67.93 | 73.22 (-3.08) |
| TA-magnitude | 71.47 | 89.01 | 62.74 | 73.49 | 83.48 | 72.43 | 75.44 (-0.86) |
| Ties-dare | 70.31 | 87.12 | 60.38 | 70.40 | 83.66 | 67.93 | 73.30 (-3.00) |
| Ties-magnitude | 71.57 | 88.93 | 62.71 | 73.49 | 84.08 | 73.26 | 75.67 (-0.63) |
| **CABS (Ours)** | **71.87** | **89.01** | **62.95** | **74.04** | **84.65** | **74.06** | **76.10 (-0.20)** |

Table 15. Impact of different n:m ratios on CABS.(sparsity=0.75)

| METHOD | ARC | Hella. | MMLU | TQA | Wino. | GSM8K | AVG |
|---|---|---|---|---|---|---|---|
| WestSeverus-7B-DPO-v2 | 71.30 | 88.26 | 63.92 | 72.72 | 83.69 | 74.27 | 75.69 |
| WildMarcoroni-Variant1-7B | 73.63 | 88.67 | 63.96 | 70.07 | 84.34 | 74.48 | 75.86 |
| Ideal Model | 73.63 | 88.67 | 63.96 | 72.72 | 84.34 | 74.48 | 76.30 |
| Task Arithmetic(Dense) | 72.52 | 89.25 | 63.39 | 74.00 | 83.46 | 73.38 | 76.02(-0.28) |
| CABS(16:64) | 72.44 | 89.08 | 63.11 | 73.38 | 84.79 | **75.11** | 76.32(+0.02) |
| CABS(32:128) | **72.92** | 88.89 | **63.50** | **74.41** | 84.63 | 74.65 | **76.50(+0.20)** |
| CABS(64:256) | 72.38 | **89.29** | 63.15 | 73.47 | **85.40** | 74.65 | 76.39(+0.09) |

## A.7. Additional Experiments on Performance Impact of Sparsification Sequence

We analyze how different sparse sequences, referring to the order in which source models (e.g., "wild" and "west") undergo sparsification during the merging process, affect the merged model's performance. In this context, "wild-first" and "west-first" indicate which model is sparsified first. Our findings, summarized in Table 16, suggest that while the order of sparsification has some impact, the effect remains relatively small.

## A.8. Rescale Experiments

In previous research, TIES utilized magnitude pruning to reduce conflicts during task vector merging but did not include a rescale step. Subsequent work on DARE introduced a two-step process: random pruning followed by rescaling with a factor of $\frac{1}{1-p}$, where $p$ is the sparsity rate. DARE demonstrated that random pruning, when combined with rescaling, could restore performance to levels comparable to the original fine-tuned models. However, DARE did not explore the effect of rescaling on magnitude-pruned task vectors.

In our experiments, we evaluated the impact of rescaling on both magnitude-based and random pruning methods across different sparsity levels. As shown in Figure 6, rescaling allows magnitude-pruned task vectors to recover performance similar to that achieved by DARE, suggesting that rescaling is a crucial step for maintaining model performance post-pruning.

These findings confirm that, with appropriate rescaling, both magnitude-based and random pruning methods can achieve near-original performance. This insight complements the primary contributions of our work by showing that magnitude pruning, which traditionally underperformed compared to random pruning in TIES, can be equally effective when combined with rescaling. Although this experiment supports the robustness of magnitude pruning under rescale conditions, it is not the main focus of our study and is therefore detailed here in the appendix.

*Table 16.* Performance comparison across different sparse sequences on LLM Leaderboard tasks.(sparsity=0.75)

| METHOD | ARC | Hella. | MMLU | TQA | Wino. | GSM8K | AVG |
|---|---|---|---|---|---|---|---|
| WestSeverus-7B-DPO-v2 | 71.30 | 88.26 | 63.92 | 72.72 | 83.69 | 74.27 | 75.69 |
| WildMarcoroni-Variant1-7B | 73.63 | 88.67 | 63.96 | 70.07 | 84.34 | 74.48 | 75.86 |
| Ideal Model | 73.63 | 88.67 | 63.96 | 72.72 | 84.34 | 74.48 | 76.30 |
| Task Arithmetic(Dense) | 72.52 | 89.25 | 63.39 | 74.00 | 83.46 | 73.38 | 76.02(-0.28) |
| CABS(16:64)-wild-first | 72.30 | 88.87 | 63.47 | 74.27 | 84.77 | 74.12 | 76.3(+0.0) |
| CABS(16:64)-west-first | 72.44 | 89.08 | 63.11 | 73.38 | 84.79 | **75.11** | 76.32(+0.02) |
| CABS(32:128)-wild-first | 72.92 | 88.89 | **63.50** | 74.41 | 84.63 | 74.65 | **76.50(+0.20)** |
| CABS(32:128)-west-first | 72.58 | 89.19 | 63.19 | 74.22 | 85.16 | 74.15 | 76.42(+0.12) |
| CABS(64:256)-wild-first | **72.87** | 89.02 | 63.43 | **74.61** | 84.37 | 73.92 | 76.37(+0.07) |
| CABS(64:256)-west-first | 72.38 | **89.29** | 63.15 | 73.47 | **85.40** | 74.65 | 76.39(+0.09) |

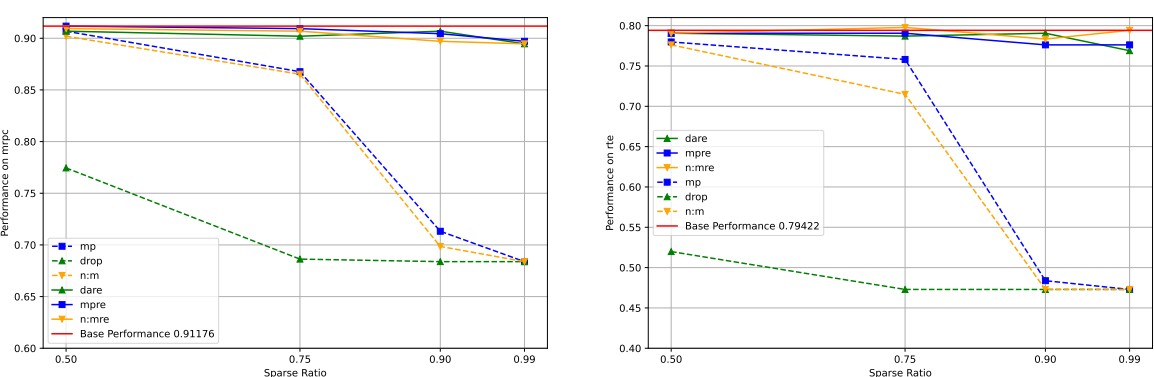

*Figure 6.* Impact of rescaling on different pruning methods across various sparsity levels. Performance is evaluated on RTE and MRPC tasks using RoBERTa. The horizontal axis represents the sparsity ratio, while the vertical axis indicates the performance of the task vectors after rescaling.

## A.9. Impact of Lambda on Performance

Figure 7 provides the average performance as a function of $\lambda$. It can be observed that within a certain range, the performance is relatively insensitive to variations in $\lambda$. This result corresponds to the performance of the CABS framework on the RTE-MRPC task. For visualization purposes, the same $\lambda$ values were used across the tasks rather than the task-specific $\lambda$ values reported in the paper. The $\lambda$ values range from 1 to 3, with a step size of 0.01, resulting in a total of 200 samples.

## A.10. Multilingual Applicability of CABS

While our primary experiments focused on English tasks to maintain comparability with prior work, we extended our evaluation to include two Korean language tasks, **kobest_copa** and **kobest_boolq** (Jang et al., 2022), to investigate the multilingual applicability of our method. These additional experiments provide insight into the performance of CABS across diverse linguistic contexts. The results are summarized in Table 17.

For these experiments, we reused the merging configuration from our previous 7B experiments to ensure consistency across evaluations and to reduce computational overhead during this phase. CABS achieves an average score of **75.41**, closely matching the ideal model's performance of **75.59** (a difference of -0.18). In comparison, the best alternative, Task Arithmetic + DARE, achieves **74.63** (-0.96), with other methods falling even further behind. These results confirm that CABS delivers competitive performance across both English and non-English tasks.

Additionally, these findings underscore the robustness of CABS in maintaining performance across multilingual benchmarks, highlighting its potential applicability to a wide range of languages and tasks. While the absolute improvement margins may vary due to upper-bound constraints imposed by the ideal model, CABS consistently demonstrates its effectiveness and adaptability across diverse settings.

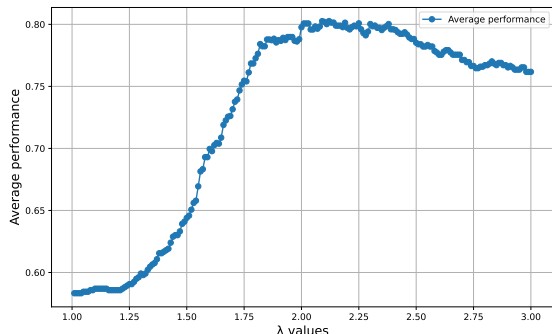

*Figure 7.* Average performance vs.lambda

*Table 17.* Performance comparison on multilingual tasks, including Korean language benchmarks.

| Model | ARC | Hella. | MMLU | TQA | Wino. | GSM8K | Kcopa | Kboolq | Avg |
|---|---|---|---|---|---|---|---|---|---|
| Mistral-7B-v0.1 | 59.98 | 83.31 | 64.16 | 42.15 | 78.37 | 37.83 | 59.00 | 62.61 | 60.93 |
| WestSeverus | 71.30 | 88.26 | 63.92 | 72.72 | 83.69 | 74.27 | 63.30 | 81.91 | 74.92 |
| WildMarcoroni | 73.63 | 88.67 | 63.96 | 70.07 | 84.34 | 74.48 | 64.80 | 82.08 | 75.25 |
| Ideal Model | 73.63 | 88.67 | 63.96 | 72.72 | 84.34 | 74.48 | 64.80 | 82.08 | 75.59 |
| TA (Dense) | 72.52 | 89.25 | 63.39 | 74.00 | 83.46 | 73.38 | 65.60 | 72.58 | 74.27 (-1.32) |
| TA + Magnitude | 71.93 | 89.32 | 63.18 | 73.85 | 84.12 | 72.22 | 64.70 | 72.86 | 74.02 (-1.57) |
| TA + DARE | 72.64 | 88.86 | 64.53 | 72.82 | 84.03 | 73.44 | 61.40 | 79.34 | 74.63 (-0.96) |
| TIES-Merging | 71.42 | 89.17 | 63.16 | 73.82 | 84.74 | 73.01 | 64.80 | 73.08 | 74.15 (-1.44) |
| TIES + DARE | 71.87 | 88.95 | 63.56 | 72.87 | 84.61 | 73.21 | 61.40 | 79.63 | 74.51 (-1.08) |
| **CABS (Ours)** | **72.92** | **88.89** | **63.50** | **74.41** | **84.63** | **74.65** | **65.10** | **79.20** | **75.41 (-0.18)** |

### A.11. Model soups experimental results

**Merging Checkpoints of the Same Task for Better Robustness.** As shown in Table 18, merging checkpoints fine-tuned on the same task improves performance, with CABS achieving the highest SST-2 accuracy of 0.9472, surpassing other methods by a notable margin (+1.49). These two checkpoints were fine-tuned for one epoch using Adam and AdamW optimizers, respectively, with a learning rate of $3 \times 10^{-5}$. The original training set was split 9:1 into a new training set and a validation set, with the validation set used as the test set. This result demonstrates the effectiveness of CABS in maintaining robustness and resolving conflicts during checkpoint merging.

### A.12. Effect of Learning Rate on Overlap Degree

We conducted additional experiments to study the effect of learning rate on the parameter overlap degree under magnitude pruning with 90% sparsity. Specifically, we fine-tuned the model using learning rates from the set

*Table 18.* Model soups experimental setup. CABS improves performance when merging checkpoints on the same tasks.

| Method | SST-2 Accuracy |
|---|---|
| Fine-tuned model1 | 0.9323 |
| Fine-tuned model2 | 0.9289 |
| Task Arithmetic | 0.9381 (+0.58) |
| +Magnitude | 0.9381 (+0.58) |
| +DARE | 0.9346 (+0.23) |
| TIES-Merging | 0.9404 (+0.81) |
| +DARE | 0.9358 (+0.35) |
| CABS(Ours) | **0.9472 (+1.49)** |

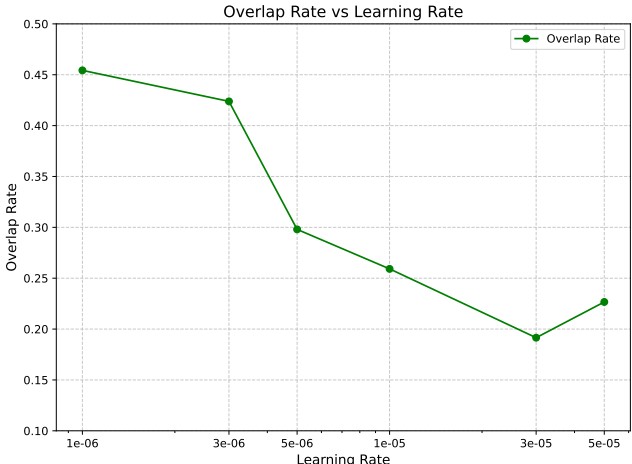

*Figure 8.* The relationship between learning rate and parameter overlap degree under magnitude pruning with 90% sparsity. Lower learning rates result in higher overlap.

$\{1e\text{-}6, 3e\text{-}6, 5e\text{-}6, 1e\text{-}5, 3e\text{-}5, 5e\text{-}5\}$ with both Adam and AdamW optimizers. After pruning, the parameter overlap degree was calculated to analyze the relationship between learning rate and parameter overlap.

Our observations, illustrated in Figure 8, show that lower learning rates lead to a higher overlap degree among parameters. This indicates that fine-tuning at lower learning rates tends to preserve shared information across tasks, even under extreme sparsity conditions. Conversely, higher learning rates result in less overlap, likely due to more significant parameter updates during optimization.

## B. Detailed Experimental Settings

### B.1. Overlap Rate Calculation

The overlap rate between two task vectors is a metric used to quantify the extent to which the same parameters are retained after pruning. This metric is particularly useful in understanding how pruning strategies impact the sharing of model parameters across different tasks, which can lead to conflicts during model merging.

The overlap rate is calculated as follows: Given two task vectors $\tau_A$ and $\tau_B$, the overlap rate is defined as the ratio of the number of shared non-zero parameters to the total number of non-zero parameters in the first task vector $\tau_A$. Mathematically, this can be expressed as:

$$\text{Overlap Rate} = \frac{|\tau_A \cap \tau_B|}{|\tau_A|}$$

where $|\tau_A \cap \tau_B|$ represents the count of non-zero parameters that are common to both vectors $\tau_A$ and $\tau_B$, and $|\tau_A|$ denotes the total count of non-zero parameters in vector $\tau_A$. This calculation shows the extent of overlap between two task vectors. A higher overlap rate means more shared parameters, increasing the potential for conflicts during model merging.

### B.2. Weight Distribution Analysis Across Layers and Sparsity Ratios

This section provides a comprehensive analysis of the heatmaps illustrating weight distributions across different layers of the model and various sparsity ratios. Figures 9-11 show the weight distribution for four representative layers: `self_attn.k_proj.weight` (layer 6), `self_attn.q_proj.weight` (layer 12), `self_attn.v_proj.weight` (layer 24), and `mlp.up_proj.weight` (layer 18) at sparsity ratios of 25%, 50%, 75%, and 90%.

These heatmaps demonstrate how increasing sparsity causes magnitude-based pruning to concentrate weights in localized regions of the parameter space. As the sparsity level increases, this clustering becomes more pronounced, especially at 75% and 90% sparsity levels, leading to potential imbalances that can degrade model performance.

The recurring pattern across all layers further highlights the significance of strategies like Balanced Sparsification (BS),

which aim to distribute weights more evenly across the model. By ensuring a more uniform distribution of the retained weights, BS helps to maintain model stability and performance after sparsification.

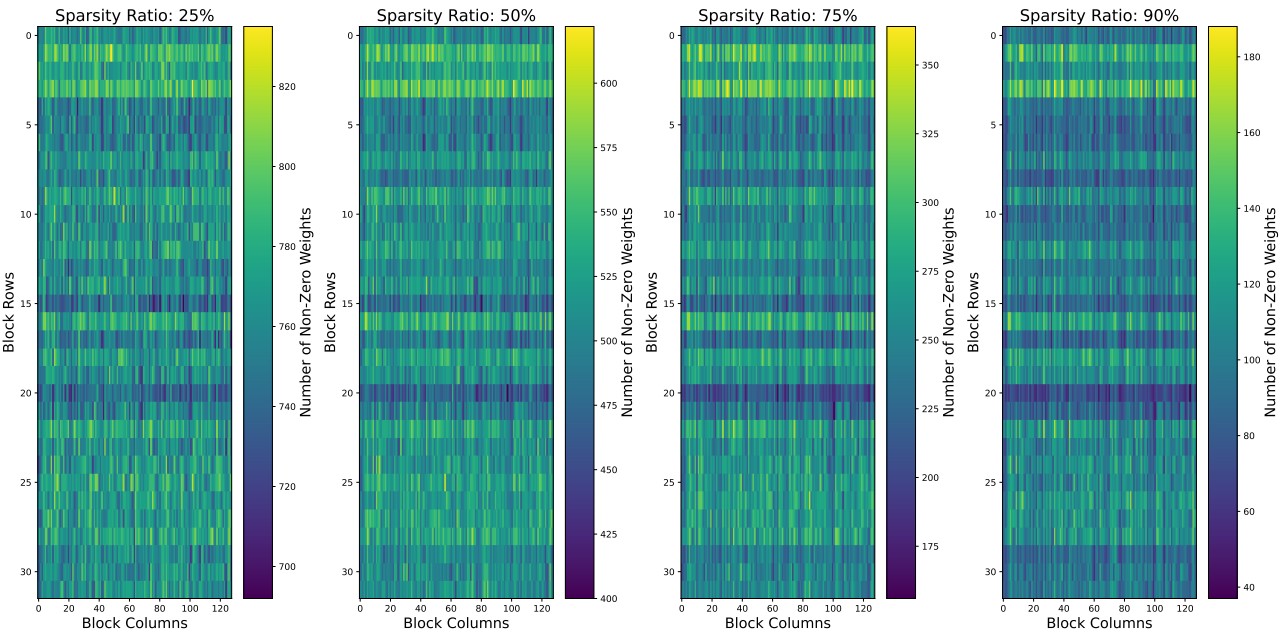

*Figure 9.* Heatmaps of weight distribution in model.layers.6.self_attn.k_proj.weight across different sparsity ratios (25%, 50%, 75%, and 90%).

### B.3. Algorithm of CABS

In this section, we present the detailed steps for both the CABS sparsity algorithm and the Low-Overlap Sparsity approach. Algorithm **??** outlines the process behind CABS, Algorithm 2 provide the detailed algorithm for Low-Overlap Sparsity designed to minimize direct conflicts during the model merging process. The algorithm sequentially applies sparsification to task vectors, ensuring that the non-overlapping portions of the task vectors are prioritized, thereby reducing overlap and conflict between different task vectors in the final merged model.

---

**Algorithm 1** CABS

**Input:** Task vectors $\tau_A, \tau_B$, base model $W_{\text{base}}$, sparsity level $n$, $m$, scaling coefficients $\lambda_A$, $\lambda_B$
**Output:** Parameters of the merged model $W_{\text{final}}$
 1: Apply n:m pruning to $\tau_A$ and compute $\text{mask}_A$

                                                             # include BS
 2: $\tau_{\text{B remaining}} = \tau_B \odot (1 - \text{mask}_A)$ to eliminate overlap with $\tau_A$     # core step of CA
 3: Apply n:m pruning to $\tau_{\text{B remaining}}$ to compute $\text{mask}_B$

                                                             # include BS
 4: Merge the pruned vectors with the base model:
$$W_{\text{final}} = W_{\text{base}} + \lambda_A \times \text{mask}_A \odot \tau_A + \lambda_B \times \text{mask}_B \odot \tau_B$$
 5: Return $W_{\text{final}}$

---

### B.4. Comparison of n:m pruning and BS

Although both n:m pruning and BS employ the same operation—selecting the top $n$ values out of $m$ consecutive weights based on magnitude—their goals and use cases differ:

- *Goal*: The primary goal of n:m pruning is to achieve model compression and acceleration by reducing computational and

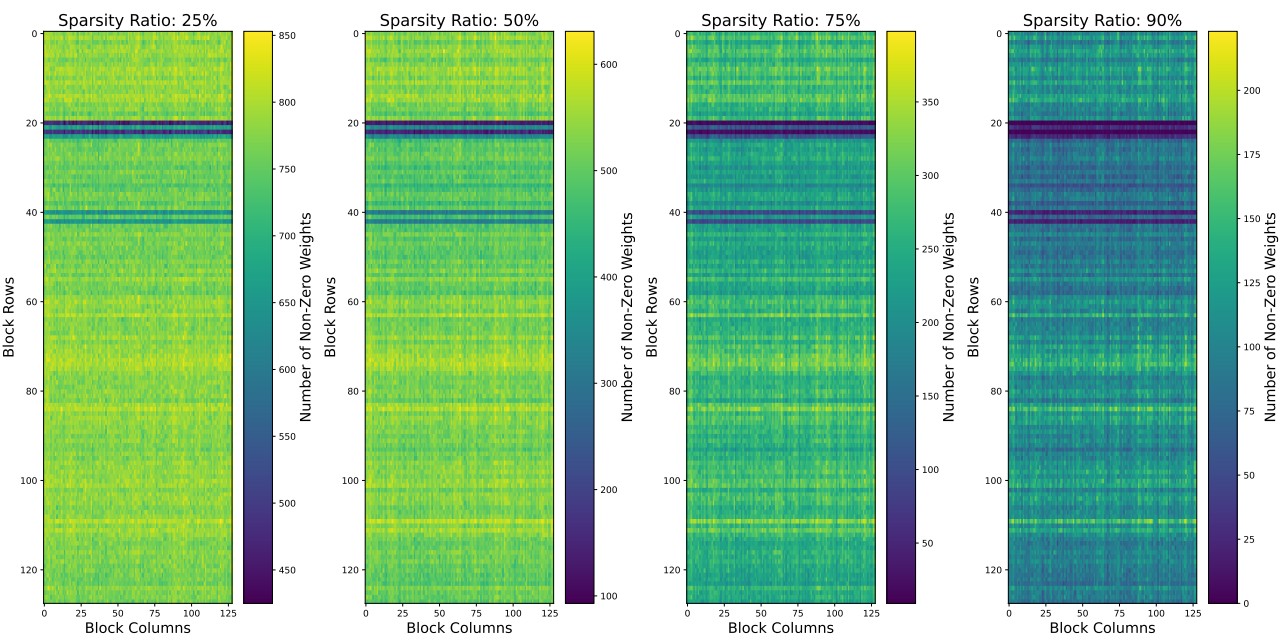

Figure 10. Heatmaps of weight distribution in model.layers.12.self_attn.q_proj.weight across different sparsity ratios (25%, 50%, 75%, and 90%).

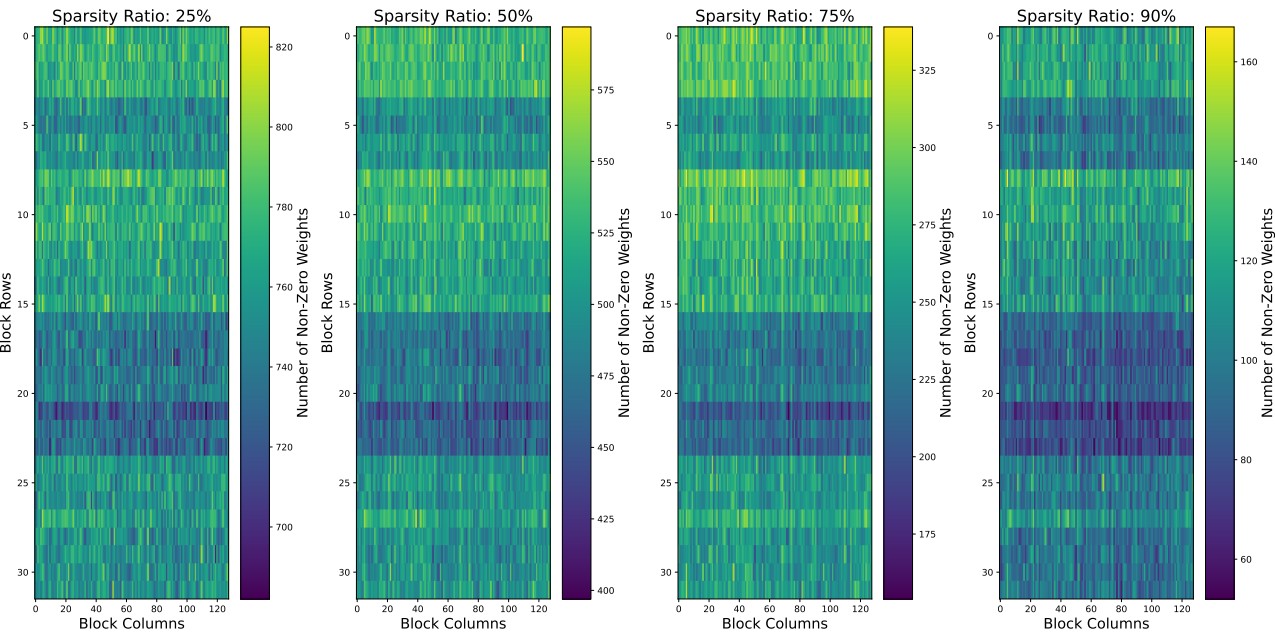

Figure 11. Heatmaps of weight distribution in model.layers.18.mlp.up_proj.weight across different sparsity ratios (25%, 50%, 75%, and 90%).

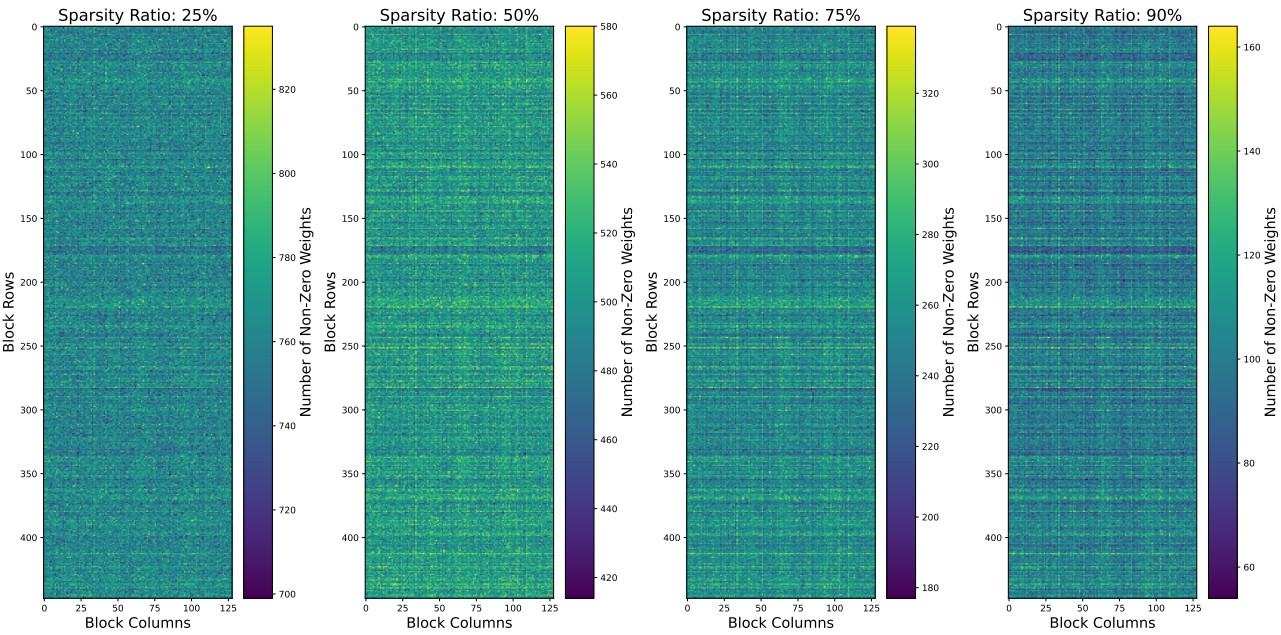

*Figure 12.* Heatmaps of weight distribution in model.layers.24.self_attn.v_proj.weight across different sparsity ratios (25%, 50%, 75%, and 90%).

---

**Algorithm 2** CABS Implementation:minimize overlap rate

**Input:** Task vectors $\tau_A, \tau_B$, base model $W_{\text{base}}$, sparsity level $n$, $m$, scaling coefficients $\lambda_A$, $\lambda_B$

**Output:** Merged model parameters $W_{\text{final}}$

1: Apply n:m pruning to $\tau_A$ and compute $\text{mask}_A$                                                        // include BS
2: Compute $\text{initial\_mask}_B = 1 - \text{mask}_A$, retaining non-overlapping regions of $\tau_B$
3: If $\text{initial\_mask}_B$ retains less than $n \div m$ of weights, update $\text{mask}_B$ by including additional weights from the overlapping region $\text{mask}_A \odot \tau_B$ until the target sparsity $n \div m$ is reached
4: Merge the pruned vectors with the base model:

$$W_{\text{final}} = W_{\text{base}} + \lambda_A \times \text{mask}_A \odot \tau_A + \lambda_B \times \text{mask}_B \odot \tau_B$$

5: Return $W_{\text{final}}$

---

memory costs. In contrast, BS is designed to maintain a balanced distribution of task vectors while minimizing conflicts between them during merging, not to merely discard unimportant weights.

- *Result*: n:m pruning is typically used for structured pruning in models, aiming to reduce inference time and memory usage. On the other hand, BS is applied specifically to task vectors. After the task vectors are merged with a base model, the resulting model remains dense, meaning that the practical computation and memory savings are not realized, but the model gains improved capacity.

- *Sparsity Ratios*: n:m pruning often uses configurations like 2:4 or 4:8, where the sparsity level is generally around 50%. In contrast, the sparsification of task vectors under BS can involve much higher sparsity levels, as can be seen in Table 15 (Appendix A.6), with configurations such as 64:256 at 75% sparsity.

- *Effectiveness*: Typically, n:m pruning yields lower performance compared to magnitude pruning in compression tasks, as the more strict uniform distribution of sparsity across blocks (e.g., every 4 weights) tends to hurt performance. However, in model merging, n:m sparsity can outperform row-wise or layer-wise magnitude pruning due to its more balanced distribution.

## B.5. Computational Overhead Analysis

This section provides a detailed analysis of the computational complexity of the CABS framework, focusing on its core components: Balanced Sparsification (BS) and Conflict-Aware (CA) pruning strategies, as well as the scalability and parallelization potential.

**Balanced Sparsification (BS)** operates efficiently by dividing each layer's parameters into small, fixed-size blocks of $m$ parameters. Within each block, the top $n$ weights are selected based on magnitude, requiring a localized sorting operation with complexity $O(m \log m)$ per block. For a layer with $N/m$ blocks, the total complexity per task vector is $O(N \log m)$, significantly more efficient than global magnitude pruning with a complexity of $O(N \log N)$. When merging $k$ task vectors, the total complexity becomes $O(kN \log m)$, making BS highly scalable for large-scale model merging.

**Conflict-Aware Sparsification (CA)** introduces minimal computational overhead by sequentially applying a mask inversion and element-wise product to ensure non-overlapping pruned regions across task vectors. These operations align with standard sparsification frameworks and maintain the same order of complexity, adding negligible cost compared to traditional methods. Combined with BS, the CA strategy ensures robust conflict resolution while maintaining computational efficiency.

**Scalability and Parallelization.** The complexity of CABS scales linearly with the number of task vectors ($k$), ensuring $O(kN \log m)$ efficiency for BS. Additionally, the block-based pruning operations in BS and the sequential processing in CA are inherently parallelizable, allowing task vector processing to occur independently across layers or blocks. This parallelization potential leverages modern hardware architectures, enabling efficient execution even for large-scale models. Without full parallelization, CABS still remains computationally efficient for real-world applications.

**Comparison and Conclusion.** Compared to traditional global magnitude pruning ($O(N \log N)$), the block-based sorting in BS ($O(N \log m)$) provides substantial computational savings. CA introduces negligible overhead, ensuring efficient and robust merging across multiple task vectors. Overall, with efficient scaling and inherent parallelization, CABS maintains a low computational overhead while effectively resolving task conflicts and ensuring balanced weight distribution, making it suitable for both small- and large-scale models.

## B.6. Memory Overhead Analysis

This section analyzes the memory overhead of CABS during the merging process and compares it to existing methods such as DARE and TIES-Merging.

**Memory Overhead of CABS.** During the merging process, CABS requires memory for storing the model parameters and two additional boolean-like masks: one to track weight usage and another to record pruning results. For a model with $N$ parameters, the memory overhead of these masks is $O(2 \cdot N \cdot 0.125 \text{ bytes})$, which is negligible compared to the memory required for storing the model parameters themselves ($O(N \cdot 2 \text{ bytes})$). As a result, the peak memory usage of CABS during the merging phase is comparable to other methods and remains efficient for large-scale models.

**Comparison with Other Methods.** DARE requires loading both source models into memory during the merging process. With lazy loading, the peak memory usage is $O(2 \cdot N \cdot 2 \text{ bytes})$, where $N$ is the number of parameters in a model. TIES-Merging, on the other hand, requires memory for all task vectors simultaneously during its election phase, resulting in $O(k \cdot N \cdot 2 \text{ bytes})$, where $k$ is the number of task vectors. However, with lazy loading, TIES-Merging can reduce its memory usage to $O(2 \cdot N \cdot 2 \text{ bytes})$, matching that of DARE. CABS achieves a similar peak memory usage as DARE and TIES-Merging with lazy loading, as the additional memory required for the two boolean masks is negligible compared to the memory needed for model parameters. This makes CABS as memory-efficient as other existing methods while offering additional robustness and performance benefits.

**Conclusion.** CABS introduces minimal additional memory overhead, as the boolean masks required for Balanced Sparsification are lightweight compared to the model parameters. Furthermore, the merging process is typically performed on CPUs, where memory constraints are less critical than on GPUs. In practice, no memory bottlenecks have been observed during experiments, confirming that CABS is memory-efficient and scalable for merging large-scale models.

## B.7. Details of Datasets and Models for LLMs

**Datasets:** Our evaluation framework comprises two benchmark suites that collectively assess a broad spectrum of language understanding, reasoning, and problem-solving capabilities.

**(1) Open LLM Leaderboard Benchmark:**

- **AI2 Reasoning Challenge**: A set of grade-school science questions designed to test fundamental reasoning skills.

- **HellaSwag**: A commonsense inference task that poses challenges for state-of-the-art models while remaining straightforward for humans (with human accuracy around 95%).

- **MMLU**: A multitask evaluation covering 57 subjects—including elementary mathematics, US history, computer science, and law—to gauge broad-domain knowledge.

- **TruthfulQA**: A benchmark that measures a model's tendency to avoid reproducing widely circulated falsehoods.

- **Winogrande**: An adversarial task based on Winograd schemas, which tests nuanced commonsense reasoning.

- **GSM8K**: A collection of grade-school math word problems that require multi-step mathematical reasoning.

**(2) Open LLM Leaderboard 2 Benchmark:**

- **IFEval**: Designed to evaluate inference capabilities across complex, varied scenarios.

- **BBH**: A subset of BIG-Bench hard tasks that challenges models with problems requiring deep reasoning.

- **MATH**: A dataset comprising challenging mathematical problems that demand multi-step, non-trivial problem solving.

- **GPQA**: A general-purpose question-answering benchmark that spans a diverse range of topics.

- **MUSR**: Focused on assessing multi-step reasoning in intricate contexts.

- **MMLU-PRO**: An advanced variant of MMLU that emphasizes professional and specialized domain knowledge.

**Models:** We evaluated two families of models corresponding to the two benchmark suites.

**(1) Open LLM Leaderboard Models:** These models are built on the `Mistral-7b-v0.1`[1] backbone and include the following fine-tuned variants:

- `WildMarcoroni-Variant1-7B`[2]

- `WestSeverus-7B-DPO-v2`[3]

**(2) Open Leaderboard 2 Models:** For the new benchmark suite, we use `Qwen/Qwen2.5-7B-Instruct`[4] as the base model, and include the following fine-tuned variants:

- `ehristoforu/fq2.5-7b-it-normalize_false`[5]

- `Tsunami-th/Tsunami-0.5-7B-Instruct`[6]

These models were selected for their robust performance across the diverse tasks and their proven utility in prior research.

---

[1] https://huggingface.co/mistral-7b-v0.1
[2] https://huggingface.co/WildMarcoroni-Variant1-7B
[3] https://huggingface.co/WestSeverus-7B-DPO-v2
[4] https://huggingface.co/Qwen/Qwen2.5-7B-Instruct
[5] https://huggingface.co/ehristoforu/fq2.5-7b-it-normalize_false
[6] https://huggingface.co/Tsunami-th/Tsunami-0.5-7B-Instruct

## B.8. Details of Datasets and Models for Small LMs

**Tasks** The GLUE benchmark includes a variety of tasks designed to evaluate different aspects of natural language understanding. For our experiments, we selected the following four tasks:

- CoLA (Corpus of Linguistic Acceptability), which evaluates the grammatical acceptability of sentences with performance measured using the Matthews Correlation Coefficient (MCC);

- SST-2 (Stanford Sentiment Treebank), a binary sentiment classification task assessing whether a sentence expresses a positive or negative sentiment, evaluated using accuracy;

- MRPC (Microsoft Research Paraphrase Corpus), a paraphrase identification task where models predict whether two sentences have the same meaning, evaluated using both accuracy and F1 score;

- RTE (Recognizing Textual Entailment), a natural language inference task where models determine whether a hypothesis is true based on a given premise, evaluated using accuracy.

- SQuAD (Stanford Question Answering Dataset): A question-answering task that evaluates models on their ability to extract precise spans of text that answer questions from a given context, measured using F1 and exact match (EM) scores.

- RACE (ReAding Comprehension from Examinations): A dataset for evaluating reading comprehension by requiring models to answer multiple-choice questions based on given passages. The dataset includes diverse linguistic phenomena, with performance measured using accuracy.

**Models** For each task, we utilized pre-trained and fine-tuned versions of RoBERTa, obtained from Hugging Face. Specifically, we used FacebookAI/roberta-base[7] as base model. textattack/roberta-base-CoLA[8], textattack/roberta-base-SST-2[9], textattack/roberta-base-MRPC[10], textattack/roberta-base-RTE[11], Riiid/kda-roberta-base-race[12] and deepset/roberta-base-squad2[13]. we also use pre-trained and fine-tuned versions of GPT-2, obtained from Hugging Face for additional experiments. Specifically, we used openai-community/gpt2[14] as the base model, tanganke/gpt2-cola[15] and tanganke/gpt2-mrpc[16].

## B.9. Evaluation Metrics

For GLUE tasks, accuracy was chosen as the uniform metric to facilitate fair comparison across tasks. While MCC is recommended for CoLA, we used accuracy to maintain consistency with other tasks. MCC typically reaches around 0.64 after fine-tuning for CoLA, whereas accuracy for other tasks often exceeds 0.9. This discrepancy makes it difficult to include MCC in an overall performance average.

For LLM Leaderboard tasks, the following metrics were used:

- **ARC**: Success rate (25-shot)

- **HellaSwag**: Accuracy (10-shot)

- **MMLU and Winogrande**: Accuracy (5-shot)

- **TruthfulQA**: Factual accuracy (0-shot)

- **GSM8K**: Success rate (5-shot)

---

[7]https://huggingface.co/FacebookAI/roberta-base
[8]https://huggingface.co/textattack/roberta-base-CoLA
[9]https://huggingface.co/textattack/roberta-base-SST-2
[10]https://huggingface.co/textattack/roberta-base-MRPC
[11]https://huggingface.co/textattack/roberta-base-RTE
[12]https://huggingface.co/Riiid/kda-roberta-base-race
[13]https://huggingface.co/deepset/roberta-base-squad2
[14]https://huggingface.co/openai-community/gpt2
[15]https://huggingface.co/tanganke/gpt2_cola
[16]https://huggingface.co/tanganke/gpt2_mrpc

These metrics provide a consistent and comparable basis for evaluating model performance across various benchmarks.

## B.10. Grid Search Details

For small-scale tasks, we performed a fine-grained $\lambda$ parameter search with an interval of 0.01 (compared to 0.1 used in previous works) to ensure fair comparisons between methods. In contrast, because of the high computational cost of large-scale experiments (e.g., with 7B models), we followed prior work by adopting a coarser grid interval of 0.1, with equal $\lambda$ values for all vectors. The impact of lambda grid intervals is discussed in Appendix A.1, showing how coarser intervals may lead to unfair comparisons by missing optimal values.

In our small-scale experiments, we employed a two-step grid search strategy to determine the optimal scaling coefficients $\lambda$ that maximizes average performance across multiple tasks.

**Grid Search Strategy** As the sparsity level increases, the range of potential optimal $\lambda$ values broadens, and performance typically follows a pattern of increasing and then decreasing with respect to $\lambda$. To address this, we adopted a two-step adaptive search strategy. First, a manual search with a 0.1 interval was performed to identify the broader region where the optimal $\lambda$ is likely to reside. Based on the results of this initial search, a more fine-grained search using a 0.01 interval was conducted, focusing on the identified region.

To further evaluate the method's ability to merge multiple task vectors ($k > 3$), additional experiments were conducted by merging four models at 90% sparsity. In these experiments, a unified $\lambda$ value was used across all task vectors, with a search interval of 0.01. This unified approach simplifies the process and mitigates the computational burden of searching for optimal $\lambda$ combinations, which would otherwise grow exponentially with the number of models $k$.

Unlike a fixed-range search, this adaptive strategy allowed us to efficiently identify the most effective scaling coefficients for each sparsity level, ensuring precise performance optimization. The performance values presented in the main text correspond to the optimal $\lambda$ values found through this two-step process.

## B.11. Guidelines and Experimental $\lambda$ Values

This section describes the guidelines for setting $\lambda$ values and presents experimental results using a unified $\lambda$ across various sparsity levels for large-scale models and across different numbers of tasks for small-scale models.

**Guidelines for Setting $\lambda$:**

- **Small-Scale Models**: A fine-grained grid search with an interval of 0.01 was used to ensure fair comparisons and avoid missing optimal values.

- **Large-Scale Models (e.g., 7B Models)**: A coarser grid search with an interval of 0.1 was adopted to reduce computational costs, consistent with prior work.

*Table 19.* Unified $\lambda$ values for large-scale models at different sparsity levels.

| Sparsity Level | Task-Arithmetic | TA-Magnitude | TA-DARE | TIES-Merging | TIES-DARE | CABS |
|---|---|---|---|---|---|---|
| 0 | 0.6 | - | - | - | - | - |
| 0.25 | - | 0.6 | 0.8* | 0.6 | 0.8* | 0.6 |
| 0.75 | - | 0.8 | 2.2* | 0.8 | 2.2* | 1.2 |
| 0.90 | - | 1.2 | 5.5* | 1.2 | 5.5* | 1.8 |

*Table 20.* Unified $\lambda$ values for small-scale models at different task numbers.

| Task Number | Task-Arithmetic | TA-Magnitude | TA-DARE | TIES-Merging | TIES-DARE | CABS |
|---|---|---|---|---|---|---|
| 4 | 0.48 | 4.61* | 1.07 | 1.88 | 5.72* | 1.74 |
| 6 | 0.49 | 5.61* | 1.04 | 1.88 | 5.41* | 1.64 |

**Notes:** For DARE-relate method, the reported $\lambda$ values (e.g., $\lambda = 2.2$ for 0.75 sparsity and $\lambda = 5.61$ for 0.90 sparsity) correspond to task vectors that have already been rescaled by a sparsity-adjusted factor (e.g., $(1/(1 - \text{sparsity}))$). However, directly using these rescaled task vectors for model merging without adjusting $\lambda$ effectively increases the step size of the $\lambda$ grid search. This results in a coarser optimization for DARE, making the comparison less fair. To address this, we ensured that the DARE method underwent a finer-grained $\lambda$ search to account for this implicit difference in grid interval and to enable a more equitable comparison with other methods.

### B.12. Hardware and Hyperparameter Configurations for Model Evaluation.

The model evaluations were performed on A100-40GB GPUs. For small-scale and discriminative tasks in GLUE, we conducted a single evaluation per model, as minimal variance was observed across repeated runs. In contrast, for generative tasks involving large models, where results can be more variable, inference was implemented via the lm-evaluation-harness v0.4.0. To ensure consistency and robustness, we performed three evaluations and reported the average outcome. As for the hyperparameters of generative LMs, we set the maximum generation token limit to 256, the temperature to 1.0 for sampling, and the maximum context length to 2048 tokens.

### B.13. Limitations and Future Work

**General Limitations.** Like other task vector-based methods, our approach is limited to models with identical architectures due to the element-wise operations used in merging model weights. This constraint restricts the generalization of the framework to models with homogeneous structures. Furthermore, reliance on manual adjustment of the parameter $\lambda$ remains a common challenge, especially for large-language models, which requires trial and error to optimize model performance.

**Limitations Specific to CABS.** CABS introduces two new hyperparameters—the sparse sequence and the n:m ratios—unique to its design, as discussed in Appendix A.7 and A.6. While these hyperparameters were not particularly sensitive in our experiments, they add complexity and increase computational cost.

**Future Work.** Several directions could help overcome these limitations. Expanding model merging techniques to include heterogeneous architectures or models trained from scratch represents a key area for future research. Additionally, improving the performance of merged models in multi-task settings—where current approaches do not yet match the performance of original single-task models—remains a priority. Automating the search for optimal hyperparameters, particularly $\lambda$, would reduce complexity and improve usability, especially in large-scale applications.

