# OpenReview forum: "CABS: Conflict-Aware and Balanced Sparsification for Enhancing Model Merging"
_ICML.cc/2025/Conference — ICML 2025 poster_

### Official Review · Reviewer_Lsxt · 2025-02-14

**Overall Recommendation:** 3

**Summary:**

This paper introduces a model merging algorithm, CABS, to address the key issues in sparsified task vectors: high parameter overlap and unbalanced weight distribution. The proposed method consists of two components: Conflict-Aware (CA) Sparsification, which sequentially prunes task vectors to minimize overlap, and Balanced Sparsification (BS), which ensures even distribution of retained parameters. The authors conduct comprehensive experiments, demonstrating that CABS outperforms existing pruning-based merging techniques and surpasses an ideal baseline in some cases.

**Claims And Evidence:**

Yes, most of the claims are well supported by evidence.

**Essential References Not Discussed:**

Most of the related work is sufficiently covered, except for a few cases (e.g., [2]). I encourage the authors to elaborate further on both [1] and [2].

**Experimental Designs Or Analyses:**

I think the experiments are comprehensive and nicely executed in general, and I particularly appreciate the experiments on large-scale language models (up to 7b). Nevertheless,
- The paper does not discuss whether CABS requires validation data for tuning the scaling coefficients, which is a critical factor in model merging methods
- The performance gap between different methods is not very large. Reporting error bars in addition to accuracy would help assess the significance of the improvements
- The paper should include comparisons with recent baselines [1,2] to strengthen its empirical claims


[1] He, Yifei, et al. "Localize-and-stitch: Efficient model merging via sparse task arithmetic." arXiv preprint arXiv:2408.13656 (2024).

[2] Wang, Ke, et al. "Localizing Task Information for Improved Model Merging and Compression." arXiv preprint arXiv:2405.07813 (2024).

**Methods And Evaluation Criteria:**

Yes

**Other Comments Or Suggestions:**

N/A

**Other Strengths And Weaknesses:**

**Strengths**:
- The proposed approach is straightforward yet effective
- The evaluation is thorough, spanning multiple tasks, models and sparsity levels

**Weaknesses**:
- Motivation: My complaint is mainly about the parameter overlap part (I think the unbalanced weight distribution is a good observation).
  - The current discussion is more or less similar to that made in [1]; the additional theoretical justifications in Sec 4.3 are shallow
  - There are a few points that I believe the authors should consider and discuss: 1) The authors claim that conflicting weight signs between task vectors lead to interference, but this is precisely what the "elect" step in TIES is designed to address. Are you suggesting that this step is ineffective? 2) [2] suggests that parameter overlap (or weight interference in their terminology) is not the primary cause of performance decline in model merging, which raises concerns about whether CABS (and similarly [1]) are targeting the right problem
- Method:
  - About "CA": The authors claim that the order of pruning does not significantly impact accuracy, but this is not necessarily true in general. A deeper analysis of the effect of ordering and some recommendations for practitioners would be a huge plus. Additionally, due to the nature of this step, there seems to be a scalability issue with respect to the number of tasks. In fact, CABS is only tested on a maximum of six tasks.
  - About "BS": While simple, the motivation behind using n:m pruning is not clearly explained, and alternative structured pruning methods should be discussed

**Questions For Authors:**

N/A

**Relation To Broader Scientific Literature:**

The paper is positioned within the broader literature on model merging. Specifically, it aims to further enhance the technique of sparsified task vectors (which could potentially mitigate the interference between different tasks) using sequential and balanced pruning.

**Theoretical Claims:**

There are some analyses in Sec 4.3, but they are pretty shallow.

---

> ### Author Rebuttal · Authors · 2025-03-31
>
> Thanks for your valuable feedback and constructive suggestions! We've provided additional experimental tables in **[Anonymous Link](https://anonymous.4open.science/r/CABS-027B/rebuttal_tables.pdf).** Below, we address your main concerns:
>
> **Q1: Comparison with recent baselines (Localize-and-Stitch [1], TALL-masks [2]).**
>
> We experimentally compared CABS with TALL-masks and Dataless Localize-and-Stitch, with our results clearly demonstrating the superior performance of CABS.
>
> For TALL-masks, please refer to our detailed response to Reviewer QfMa (Q5) for a comprehensive comparison.
>
> For Dataless Localize-and-Stitch, experimental results are presented in Tables 4, 5, and 6 of the anonymous link. When merging 2, 4, or 6 models, its performance is often close to the base model, indicating limited effectiveness.  We found it essentially equivalent to simplified TIES-Merging (fixed λ=1, no 'elect').
>
> This setting works reasonably well in the original setup, where 12 models are merged using 5–10% sparsity, so that λ = 1 happens to act as a good approximation (e.g., 5-10% × 12 ≈ 0.6-1.2). However, when merging a smaller or larger number of models, such a fixed λ becomes suboptimal.
>
> If desired, we are happy to include a λ-tuned version of Dataless L-and-S for fairer comparison.
>
> **Q2:Effectiveness of "elect" step in TIES.**
>
> We appreciate the valuable ideas introduced by TIES-Merging, particularly its insight and solution regarding parameter redundancy and conflict. Inspired by TIES-Merging, CABS offers a more robust alternative.
>
> Specifically, the “elect” step in TIES is designed to resolve conflicting signs between task vectors and can indeed be helpful in certain cases. For example, as shown in Table 17 of our appendix, TIES-Merging is the best-performing baseline after CABS. However, its effectiveness is inconsistent. The TIES paper itself (Appendix B.1, Table 7) shows that re-estimating the sign vector via few-shot training improves performance, indicating that its elect strategy may be suboptimal.
>
> Our experiments show similar trends. As seen in Table 1, the performance gap between Task Arithmetic+Magnitude and TIES-Merging is often small despite the difference being the elect step, and in some cases, TIES-Merging even underperforms.
>
> One possible explanation is the significant imbalance in average magnitudes among task vectors—up to a 10× difference as illustrated in Table 8 of the anonymous link. In such cases, the sign vector may be dominated by the task with the largest magnitude, resulting in biased merged results.
>
> Overall, while TIES-Merging laid important groundwork in addressing parameter conflicts, CABS advances this direction by providing a more robust and effective solution.
>
> **Q3: Are parameter interference the right problem？**
>
> As noted in our response to Reviewer QfMa (Q5), CABS and TALL-masks operate under fundamentally different paradigms.
>
> While TALL-masks separates tasks during inference and does not explicitly address parameter interference, it incorporates TIES-Merging as the first stage in its pipeline. This design choice implicitly acknowledges parameter interference as a key factor.
>
> In such settings—where task identity is unavailable and generalization is required—parameter conflict becomes an inevitable challenge for training-free model merging methods.
>
> **Q4: motivation for choosing n:m pruning.**
>
> We choose n:m pruning for two reasons: (1) Prior methods, like row-wise (topk), already outperform layer-wise pruning, making it natural to explore more structured forms, like block-wise (n:m), to further improve merging performance. (2) n:m pruning operates at the finest granularity (weight level), is simple to apply, and requires no changes to model architecture.
>
> Other pruning techniques are less suitable for training-free merging. Coarse-grained methods (e.g., head pruning) cannot remove redundant weights or resolve conflicts within components. Sparse fine-tuning and similar methods require training, limiting their practicality.
>
> **Q5: Potential scalability concerns for CA.**
>
> Please refer to our detailed response to Reviewer zNot (Q4).
>
> **Q6: Analysis on the impact of pruning order.**
>
> Please refer to our detailed response to Reviewer QfMa (Q6).
>
> Building on our new findings on task vector magnitudes (Table 8 in the anonymous link), we suggest that placing task vectors with smaller magnitudes earlier in the pruning sequence—or assigning them larger λ—may help improve balance. This could be a promising future direction, as exhaustive search is infeasible in LLM merging.
>
> **Q7: necessary of Validation data usage for tuning λ.**
>
> Like TA, TIES, and DARE, we use validation data for tuning λ. The specific values are provided in Appendix B.11.
>
> **Q8: Reporting statistical significance with confidence intervals.**
>
> Please refer to our detailed response to Reviewer QfMa (Q4).
>
> We hope these responses address your concerns. Please feel free to let us know if anything remains unclear.

---

> > ### Comment · Reviewer_Lsxt · 2025-04-02
> >
> > Thanks for the response. In light of the new experiments conducted, I am raising my score to 3. I do not increase further because I generally feel that the work is solid but not particularly exciting.

---

> > > ### Author Response · Authors · 2025-04-02
> > >
> > > Thank you for your thoughtful review and for taking the time to reassess our work. We are glad we could address your concerns and sincerely appreciate your updated score!

---

### Official Review · Reviewer_zNot · 2025-03-11

**Overall Recommendation:** 2

**Summary:**

This paper introduces a task-vector-based model merging method, CABS. The authors attribute the performance degradation of model merging to: (1) high parameter overlap, and (2) unbalanced weight distribution. The proposed CABS mainly contains two modules, CA and BS. Between them, CA aims to eliminate parameter overlap and BS is introduced to guarantee a more even weight distribution. Experiments on both LLMs and small language models (SLMs) are conducted to validate the performance of the proposed method.

**Claims And Evidence:**

The authors attribute the performance degradation of model merging to high parameter overlap, and unbalanced weight distribution. I agree with the viewpoint that parameter overlap degrades the performance, which is intuitive. As to parameter overlap, the authors demonstrate that the pruning based on magnitude results in unbalanced distribution of weights. However, from my point of view, I believe that the success of DARE is because of its rescaling process. I guess that if magnitude-based pruning is properly rescaled, it can achieve the performance close to DARE. As a result, I think that the relationship between weight distribution and the merged model’s performance needs to be further proved.

**Essential References Not Discussed:**

N/A

**Experimental Designs Or Analyses:**

Yes, I checked the experiments in Section 5 and the additional results in Appendix A. I have several questions:
(1)	The selected LLMs have similar performance on the target tasks. For example, the performance difference of WestSeverus and WildMarcoroni is below 1% on most tasks in Table 3. However, from my point of view, model merging targets at replacing multi-task learning by merging expert models on different kinds of tasks. Say, merging LLMs finetuned on code or math tasks. Merging models with very similar performance may reduce the soundness of the method.
(2)	For SLMs, the authors selected 4 tasks from GLUE and 2 other tasks in Table 2. However, I checked the huggingface links for roberta models utilized by the authors and find that there are many other weights that can be used, including roberta models finetuned on other tasks from GLUE. Similarly, in Table 11, only two GPT-2 models are utilized for merging but the original setting for this benchmark is merging 7 GPT-2 models finetuned on different tasks. Providing experimental results for merging more models make the proposed method more convincing.

**Methods And Evaluation Criteria:**

Yes. The proposed method is simple and provides an interesting perspective for model merging. However, some of its claims and experiments are not that convincing to me.

**Other Comments Or Suggestions:**

(1)	In Line 1120, there is a ref error.
(2)	In Figure 6, the font sizes are a bit too small to read.

**Other Strengths And Weaknesses:**

Strengths:
(1) Simple and intuitive idea.
(2) Detailed proofs.
(3) Comprehensive experimental results.
(4) Ablation studies provided.

Weaknesses (both previous-mentioned):
(1) The relationship between weight distribution and the merged model’s performance is not that convincing to me.
(2) Some experimental settings could be broadened.

**Questions For Authors:**

Overall, I think the proposed method is meaningful, but it may lack a motivation strong enough and convincing experimental results. This is why I submit a negative overall recommendation. However, if the authors could:
(1)	further prove the relationship between weight distribution and performance;
(2)	expand the experimental settings.
I would carefully consider raising my rating score.

**Relation To Broader Scientific Literature:**

This paper attributes the performance degradation to high parameter overlap and unbalanced weight distribution. Previous studies usually use pre-processing to reduce the parameter overlap or interference. In this paper, the proposed CABS applies pre-computed masks to avoid the overlap.

**Theoretical Claims:**

Yes. I checked Eq. 2 to Eq. 9. No obvious errors are found.

---

> ### Author Rebuttal · Authors · 2025-03-31
>
> Thanks for your valuable feedback and constructive suggestions! We've provided additional experimental tables in **[Anonymous Link](https://anonymous.4open.science/r/CABS-027B/rebuttal_tables.pdf).** Below, we address your main concerns:
>
> **Q1: Whether the observed performance gap is due to lack of rescaling in magnitude-based pruning.**
>
> We agree that rescaling is a key factor contributing to DARE’s effectiveness and could enhance the performance of magnitude-based pruning methods. In our experiments, all magnitude-based pruning methods were equipped with rescaling. The λ values used for merging (e.g., λ=1.88 for 4-model TIES-Merging) are detailed in Appendix B.11, confirming that proper rescaling was employed. Additionally, we also conducted rescaling experiments for magnitude-based pruning methods, which are reported in Appendix A.8.
>
> Notably, as illustrated in Table 1, even when rescaling is applied, magnitude-based pruning can sometimes impair the model merging performance. The comparison in Table 1 underscores that factors beyond rescaling - such as weight distribution - are also important for the merging performance, as further discussed in our response to Q2.
>
> **Q2: The impact of unbalanced weight distribution on merged model performance needs clearer evidence.**
>
> We provide supporting evidence in Table 4 in the main text. It compares different sparsification strategies for model merging: Task Arithmetic + Magnitude (layer-wise), + Magnitude (row-wise), and + BS (block-wise). As the pruning becomes more structured and the weight distribution more balanced, performance improves progressively from 80.38 to 80.61 to 81.30.
>
> From a theoretical perspective, weight imbalance can amplify task interference during merging. Since the merged model is typically rescaled by a λ, if weights are highly concentrated in specific regions—as often caused by magnitude-based pruning—this rescaling will disproportionately amplify those regions, worsening cross-task interference.
>
> While weight distribution may not be the sole cause, consistent improvements across balanced pruning provide strong empirical support. We acknowledge that deeper theoretical understanding, especially of activation dynamics, would be valuable. While our current work focuses on developing an effective, training-free merging method, we see this as a promising direction for future research.
>
> **Q3: The LLMs used for merging are too similar in performance; more diverse tasks are expected.**
>
> To address the concern about task diversity, we additionally conducted an experiment merging a Mistral model fine-tuned on instruction-following with another fine-tuned on mathematical tasks. As shown in Table 2 in anonymous link, our method still outperforms TaskArithmetic by +0.79, achieving SOTA performance in this heterogeneous setting and demonstrating robust cross-task merging capability.
>
> We also would like to clarify that, merging models fine-tuned on similar tasks is a standard and widely adopted setup in model merging, especially under the model soup paradigm. Our large-model experiments follow DARE’s protocol, which focuses on improving performance or generalization within similar task domains.
>
> This setup is also practically relevant: the Open LLM Leaderboard—one of the most active community benchmarks—frequently features submissions involving merges of LLMs fine-tuned on similar tasks. Using our method, we constructed four merged models and occupied the top four positions among all models with fewer than 8B parameters at the submission time (see Table 1 in the anonymous link or the official leaderboard).
>
> **Q4: The number of tasks/models in SLM experiments is limited; merging more models is recommended.**
>
> We appreciate the suggestion and have added an experiment merging all 7 GPT-2 models from FusionBench, following its official setup (see Table 7 in the anonymous link).
>
> While CABS still achieves the best performance among all methods, the overall gains are notably smaller than in our 2- or 6-model merging settings. In fact, all methods—including CABS—suffer significant performance drops when merging more models, especially on some tasks.
>
> We investigated this and found that the task vectors have highly imbalanced average weight magnitudes—up to a 10× difference across tasks (see Table 8 in the anonymous link). Since model merging uses a shared scaling factor (λ), tasks with small-magnitude vectors (e.g., RTE, MRPC, CoLA) are overwhelmed by stronger ones (e.g., QQP, MNLI), resulting in near-random performance on weaker tasks.
>
> This suggests that blindly merging too many heterogeneous task vectors with vastly different magnitudes introduces severe imbalance and can degrade model utility. While CABS remains robust, merging fewer well-aligned models appears more practical and effective under current frameworks.
>
> We hope our responses have addressed your concerns. Please let us know if there are any remaining questions or clarifications we can provide.

---

### Official Review · Reviewer_8hif · 2025-03-13

**Overall Recommendation:** 2

**Summary:**

The paper presens CABS, a method for pruning and merging different task vectors, seemingly resolving conflicts between different tasks. As the authors argue, conflicts can arise due to parameter overlap and unblanaced weight distribution - CABS is intended to address these issues. To that end, disjoint pruning masks are used, while the latter problem is resolved using n:m pruning. Experiments are conducted to show the effectiveness of the method.

## Update after rebuttal
I thank the authors for their rebuttal. However, since I find some of their arguments not really convincing, I intend to keep my score. I am not convinced that pruning, originally a method to compress models, is here a better idea than otherwise merging the task vectors. Especially the statement "when task vectors are highly similar, their directions are nearly aligned, causing their linear combination to span only a narrow subspace" does not really resonate with me. While this statement as is is untrue (pick any vector, add eps*canonical vectors to it, giving you a full dim subspace and all vectors are arbitrarily close", I am not convinced why this should be an argument in the first place. Overall, I think the work needs revision to resolve also the causation/correlation issue I stated.

**Claims And Evidence:**

- While I like the storyline of the paper, I am not convinced that the experiments show that high parameter overlap and unbalanced weight distribution are the issues in model merging, the experiments presented are not causal, but rather show correlation. For example, the overlap is larger when doing magnitude pruning, instead of random pruning, cf. Figure 2. Why more overlap should lead to worse performance is not clear - I understand the arguments when e.g. one task is positive at a certain weight, the other one negative, but why would pruning one then be any better? I believe the numbers, but I do not think that the experiments show that these are truly the issues.
- When merging two task vectors, why are you pruning both to then build up a linear combination - why doesn't it work to build a new task vectors that is accounts for both tasks, e.g. on a calibration dataset? I find the strategy a bit convoluted.
- When pruning task A, then using the inverse of the mask to restrict the weights of task B, which are then in turn pruned, do I not necessarily prune either A or B too much? In the end, this guarantees that the task vectors are orthogonal, which seems to be much much more than what is needed - if two tasks are fairly similar, why force them to orthogonality?
- From Eq. 7 to Eq. 9 it is derived that the "decoupling" in the norm "ensures that adjust $\lambda_A$ affects only the contribution of $\tau_A$", but this is not clear to me. This statement holds in Equation 7 already, why would the norm be needed here?

**Essential References Not Discussed:**

None that I am aware of.

**Experimental Designs Or Analyses:**

See above.

**Methods And Evaluation Criteria:**

The experimental design overall seems sound, however I am not entirely sure which sparsities are used, e.g. in Section 5.2.

**Other Comments Or Suggestions:**

- The running title has not been updated and is still the default one.
- I find it a bit odd to cite Liang et al. when introducing magnitude pruning in line 34. Why would you cite a 2021 paper to introduce one of the most basic pruning methods? Later, Zhu & Gupta are cited (line 156), which seems more appropriate, but still not really fitting - they introduced basically a pruning schedule which is based on magnitude pruning.
- There are some typos in the paper, e.g. "Remaining" in line 83 should be lowercase, "compare to" should be "compared to" in the caption of Figure 3.
- Here, sparsity seems to be used in the sense of density. I see this from time to time in other papers, but it is not the correct usage in my opinion. A sparsity of 80% means that 80% of the parameters are zero. A density of 80% would mean that 20% of the parameters are zero.

**Other Strengths And Weaknesses:**

- The derivation in lines 259-274 is trivial and does add little value, I would have preferred to use this space to explain why "these sparse updates are nearly lossless in retaining task-specific information, as simple rescaling compensates for pruning-induced changes". This statement may hold for very little sparsities, but otherwise this will not be the case.

**Questions For Authors:**

- What is used as the calibration dataset for SparseGPT and Wanda? Did I miss this or is this nowhere stated?

**Relation To Broader Scientific Literature:**

The authors do a good job of relating their work to the broader literature.

**Theoretical Claims:**

N/A

---

> ### Author Rebuttal · Authors · 2025-03-31
>
> Thanks for your valuable feedback and constructive suggestions! Below, we address your main concerns:
>
>  **Q1: Causality vs. Correlation in Experiments**
>
> We acknowledge the reviewer’s distinction between correlation and causation. While strict causal proof is difficult, our experiments go beyond correlation. In Figure 5, by fixing sparsity and method and varying only the overlap rate, we observe performance drop—indicating a causal link.
>
> Table 4 shows that more structured pruning improves balance and performance (80.38 → 80.61 → 81.30). While full causality is not isolated, consistent trends across models highlight the practical value of addressing overlap and imbalance in training-free model merging.
>
> **Q2: why would pruning one then be any better?**
>
> Prior work shows that pruning improves model merging. TIES-Merging demonstrates that conflicting signs between task vectors cause interference, and pruning one side improves performance. DARE further shows that task vectors are highly redundant—keeping only ~5% of weights with proper rescaling still maintains near-lossless performance.
>
> Unlike standard model pruning, task vector pruning is not for compression, but for reducing conflict and removing redundant updates that may interfere with other tasks—making it a strategic, empirically supported choice.
>
> **Q3: Why merge Task Vectors Instead of Learning a Joint Vector?**
>
> Task-vector-based merging aims to reuse independently fine-tuned models in a training-free manner, without access to original task data. In contrast, learning a joint vector requires access to all task data and additional training, which significantly increases computational cost.
>
> CABS builds on TIES-Merging and DARE, proposing structured and conflict-aware pruning to enable efficient, training-free model merging.
>
> **Q4: Does CA over-prune task vectors by enforcing disjoint masks?**
>
> CA does not cause excessive pruning. In many cases, task vectors already reach 90% sparsity without CA. At the same sparsity level, CA improves merging by allocating disjoint weights, reducing interference while retaining essential updates.
>
> **Q5: If two tasks are fairly similar, why force them to orthogonality?**
>
> In our experiments with similar models—such as the model soup setting in Table 3 and Table 17 of the main paper, and Table 1 in **[Anonymous Link](https://anonymous.4open.science/r/CABS-027B/rebuttal_tables.pdf)**—we consistently observe that enforcing orthogonality via CABS still improves merging performance.
>
> One geometric explanation is that, when task vectors are highly similar, their directions are nearly aligned, causing their linear combination to span only a narrow subspace. In contrast, enforcing orthogonality increases the span of the merged space. This broader coverage may facilitate better exploration of the solution space, enabling access to lower-loss regions.
>
> Thus, even for similar tasks, CABS effectively improves expressiveness and reduces redundancy during the merging process.
>
> **Q6: Clarifying Eq. 7–9**
>
> Equation 7 defines a linear combination of two task vectors, but does not reveal how adjusting one scaling factor (e.g., λ_B) may interfere with the other task vector’s contribution in the resulting update.
>
> For example, if τ_A = −τ_B, even small changes in λ_B can diminish the effect of τ_A, indicating strong coupling between the scaling factors. This illustrates that without further analysis, the two task vectors' contributions cannot be assumed independent.
>
> The squared Frobenius norm in Eq. 8 explicitly captures this interaction via the inner product ⟨τ_A, τ_B⟩_F. As shown in Eq. 9, the cross term vanishes only when τ_A and τ_B are orthogonal.
>
> This ensures that each λ independently controls its corresponding task vector’s contribution without unintended interference. Hence, the norm-based formulation is essential to formalize the decoupling effect of orthogonality.
>
> **Q7: Density vs Sparsity**
>
> We consistently use "sparsity" to refer to the percentage of pruned (zero) parameters—e.g., 90% sparsity means 90% of weights are zero. We do not use "density" in the paper, so there should be no ambiguity.
>
> That said, we acknowledge that Table 2 does not explicitly mention the sparsity setting. However, the table references specific results in Tables 1, 8, and 9, where the sparsity level (90%) is clearly stated. We will revise Table 2 to include this information. We hope this clarification resolves the misunderstanding.
>
> **Q8: Why does rescaling work?**
>
> Rescaling is effective because task vectors are highly redundant—shown by DARE and our Figure 6 (Appendix A.8). Pruning up to 90%, with rescaling, yields near-lossless performance. Even at 99%, the drop is minimal, showing strong robustness.
>
> **Q9: The calibration dataset for SparseGPT and Wanda**
>
> We use C4 as the calibration dataset, following the standard setup in SparseGPT.
>
> We hope these responses address your concerns. Please feel free to let us know if anything remains unclear.

---

### Official Review · Reviewer_QfMa · 2025-03-13

**Overall Recommendation:** 3

**Summary:**

Authors propose a novel methodology, Conflict Aware Balanced Sparsification (CABS), for model merging based on task vectors. Previous work has shown that sparsifying task vectors before merging typically yields better performance for merged model. Authors identify two main issues:
- High Parameter Overlap: Retained weights across task vectors overlap significantly, causing conflicts during merging.
- Unbalanced Weight Distribution: Sparsification concentrates weights in specific regions, amplifying imbalances during merging and degrading performance
Authors propose to resolve this by proposing:
A) Conflict Aware pruning: Prunes task vectors sequentially, masking already retained parameters to eliminate overlap.
B) Balanced Sparsification: Structured n:m pruning of task vectors to enforce uniform pruning of task vectors.

Authors evaluate their approach on large LM (Mistral-7B) and small LMs (RoBERTa and GPT-2). Experiments show CABS generally outperforms existing methods when merging 2, 4, and 6 tasks language modelling tasks.

**Claims And Evidence:**

Authors run ablation studies to verify most of their claims.
Authors claim that they are first to introduce an "ideal" baseline and CABS outperforms this ideal baseline.
- First, the ideal baseline authors proposed is to compare the performance of the merged model to the finetuned expert. This is pretty well known and has been used as a baseline in this [1].
- Second, The claim that CABS outperforms this baseline is misleading because only setting is in Table 3 with sparsity =0.75. CABS has average accuracy of 76.5% and "ideal" model has average accuracy of 76.3%. These results are within the margin of errors and I would encourage the authors to withdraw this claim.

[1] Ke Wang, Nikolaos Dimitriadis, Guillermo Ortiz-Jiménez, François Fleuret, and Pascal Frossard. 2024. Localizing task information for improved model merging and compression. In Proceedings of the 41st International Conference on Machine Learning (ICML'24), Vol. 235. JMLR.org, Article 2057, 50268–50287.

**Essential References Not Discussed:**

Authors should include discussion / comparison with [1]. This is published work that also appears to improve on the baselines considered in this work and seems pretty related.

[1] Ke Wang, Nikolaos Dimitriadis, Guillermo Ortiz-Jiménez, François Fleuret, and Pascal Frossard. 2024. Localizing task information for improved model merging and compression. In Proceedings of the 41st International Conference on Machine Learning (ICML'24), Vol. 235. JMLR.org, Article 2057, 50268–50287.

**Experimental Designs Or Analyses:**

I checked the experimental design for evaluating the CABS and the ablation studies.

1. This work lack of confidence intervals in the evaluation. It appears most of the work is produced on a single seed. It'll be useful to report margin of error which could boost significance of the work.
2. Authors argue that overlapping task vectors typically lead to performance degradation. I can imagine tasks where the data distributions are very similar where, having overlapping task vectors can indeed boost the performance. Thus, this analyses is not very convincing.

**Methods And Evaluation Criteria:**

For small LMs, authors tested on GLUE tasks (CoLA, SST-2, MRPC, RTE, SQuAD, RACE), arbitrarily chosing 2, 4, and 6 tasks to merge.
For large LMs, authors tested on LLM Leaderboard tasks (ARC, HellaSwag, MMLU, TruthfulQA, Winogrande, GSM8K)

These benchmarks are standard and make sense for text data modalities. However authors have not run experiments on other data modalities, such as vision tasks.

**Other Comments Or Suggestions:**

N/A

**Other Strengths And Weaknesses:**

Strengths:

- Novel merging approach that improves the performance of model merging in evaluated tasks
- Strong ablation studies including studying the role of sparsity rate, overlap rate, pruning approaches, and general experimental setup.
- Paper is well structured.

Weakness:
- Only text data modality is considered. In this literature, it is common practice to also evaluate results on vision data modality.
- Claim about comparison to "ideal" baseline are not justified.

**Questions For Authors:**

1. For CA approach: does the sequential nature of masking induce an implicit ordering on the "importance" of tasks? Have you tried shuffling this order and seen any difference in the performance?

2. Figure 5 suggests that perhaps small overlap rate (20%) can be beneficial for model performance. Can you include the performance of CABS with 0% overlap rate on that plot? it'll be useful to compare the performance with varying overlap rates.

**Relation To Broader Scientific Literature:**

Prior work has shown that sparsifying task vectors can lead to improved performance in model merging. In this work authors suggest that  removing overlaps in the task vectors + using structured n:m sparsity can improve the performance of model merging.

**Theoretical Claims:**

There are no significant theoretical claims int he paper. Authors show that CA procedure produces non-overlapping task vectors leading to orthogonality. Thus leading to no interference. This analysis is pretty straightforward.

---

> ### Author Rebuttal · Authors · 2025-03-31
>
> Thanks for your valuable feedback and constructive suggestions! We've provided additional experimental tables in **[Anonymous Link](https://anonymous.4open.science/r/CABS-027B/rebuttal_tables.pdf).** Below, we address your main concerns:
>
> **Q1: Inclusion of experiments on vision modality.**
>
> We added experiments on vision tasks (ViT-B-32 on DTD and EuroSAT, see Table 2 in the link). CABS surpasses Task Arithmetic by +2.20% average accuracy, confirming its effectiveness on vision tasks.
>
> **Q2: Justification of claims regarding outperforming the "ideal model".**
>
> We agree that this claim should be more cautiously presented. We will revise the paper to clarify the contribution statement and the discussion of Table 3 results, adding appropriate qualifiers to avoid overstatement.
>
> Empirically, the claim is supported by consistent results across multiple settings (Tables 3, 11, 12, 14, and 15), including both Mistral and GPT-2 models and sparsity levels of 0.25 and 0.75 and 0.90. In particular, the improvement in GPT-2 experiments (Table 11) is free from randomness, as both the method (CABS) and the evaluation tasks (classification) are deterministic. As noted in Q4, confidence intervals in the large-model experiments further strengthen the reliability of the observed gains.
>
> **Q3: Clarification about the novelty of the "ideal model".**
>
> We agree that comparing to fine-tuned models is common for SLM, where the "ideal baseline" is equivalent to the ''fine-tuned models''. However, in LLM merging (e.g., multi-task LLMs), this type of upper-bound baseline is not used in prior large-model merging works such as DARE. We acknowledge that this distinction was not clearly explained and will revise the text accordingly.
>
> **Q4: Reporting confidence intervals.**
>
> Our magnitude-based merging method is deterministic for classification tasks, producing consistent results across runs. This aligns with common practice in model merging literature, where confidence intervals are often omitted. Still, for LLM evaluations, we conducted 10 runs:
> |Method|Mean Score|95% Confidence Interval|
> |-|-|-|
> |TIES-DARE|0.7606|(0.7597, 0.7614)|
> |CABS|0.7647|(0.7641, 0.7653)|
>
> These results confirm the statistical significance of our improvements.
>
> **Q5: Discussion of related work TALL-masks.**
>
> Compared to TALL-masks, CABS distinguishes itself in both design and applicability:
>
> - TALL-masks assumes task id is available at inference, loading tuned task-specific masks—analogous to utilizing separate LoRA adapters—thus achieving near-lossless performance but requiring task-specific inference and added complexity.
>
> - TALL-masks also involves tuning the mask sparsity factor for each task and scaling coefficients λ. While effective on small models, it has not been evaluated at LLM scale. In contrast, CABS only tunes λ and has demonstrated strong scalability: on the Open LLM Leaderboard, CABS occupied the top four positions among all sub-8B models at submission time (Table 1 in the link).
>
> - TALL-masks produces a base model, a merged task vector, and k task-specific masks, resulting in >2× storage overhead. In contrast, CABS yields a single, compact model for multi-task inference.
>
> Given these differences, direct comparison with TALL-masks is inappropriate. However, its merged variant Consensus TA is a fair baseline. We have added comparisons on merging 2,4,6 models (Table 4,5,6 in the anonymous link, Mask sparsity factor =0.5), with our results clearly demonstrating the superior performance of CABS. A discussion will be added in the Related Work section.
>
> **Q6: Effect of task ordering in CA.**
>
> We tested shuffled task orders explicitly (SLM: Table 1 in the main text; LLM: Appendix A.9). We further added results on merging six models with shuffled orders (Table 5 in the anonymous link). Results show that task ordering has limited impact relative to overall gains, demonstrating the robustness of the CA strategy.
>
> **Q7: Impact of overlap rates.**
>
> As discussed in the paper, the main issue is high overlap rather than all overlap. We agree some overlap can be beneficial—e.g., Figure 5 shows 20% sparsity achieving strong performance. However, determining the optimal overlap level introduces additional complexity. Considering the efficiency and generalizability, we adopt the current CA strategy.
>
> Figure 5 is intended to analyze the effect of varying overlap rates within a fixed method (DARE), and already includes the result of applying CA with DARE. Adding CABS—which uses a different method (BS)—would conflate method differences with overlap effects.
>
> Instead, we have updated Table 4 in the main paper (see Table 9 in the anonymous link) to enable a clearer comparison between BS and DARE under the same sparsity level.
>
> **Q8: Effectiveness on similar tasks.**
>
> We discuss this in our response to Reviewer 8hif (Q5).
>
> We hope our responses have addressed your concerns. Please let us know if there are any remaining questions or clarifications we can provide.

---

> > ### Comment · Reviewer_QfMa · 2025-04-03
> >
> > I thank the authors for their rebuttal and answering my questions. Overall, the contributions in the paper are interesting and I am still leaning towards an accept.

---

### Decision · Program_Chairs · 2025-05-01

**Decision:**

Accept (poster)

**Comment:**

This paper presents an approach that uses task vectors to merge models, enabling the creation of a single multitask model without requiring retraining.  The authors avoid conflicts between task vectors suffered by prior work by using masking to help ensure each task vector contains distinct, non-overlapping parameters.  Reviewers are split on the outcome of this paper, with arguments in favor citing the novelty of the proposed approach and the strong empirical results, with arguments against requesting more evidence that the cited issues with prior work (e.g., high parameter overlap) are casually related with the performance drops or additional experiments on more datasets. While the former was not completely addressed, the additional experiments were provided and satisfied reviewers who updated their reviews post-rebuttal.  As such, while there are clear concerns about the current paper, the ACs find they are not sufficient to argue for rejection, and the authors are encouraged to address issues raised about their paper in their revision.